# Evolution of lysine-specific demethylase 1 and REST corepressor gene families and their molecular interaction

Montserrat Olivares-Costa[1,2,9], Gianluca Merello Oyarzún[1,3,4,9], Daniel Verbel-Vergara[1,9], Marcela P. González[1], Duxan Arancibia[5], María E. Andrés [1✉] & Juan C. Opazo [6,7,8✉]

Lysine-specific demethylase 1A (LSD1) binds to the REST corepressor (RCOR) protein family of corepressors to erase transcriptionally active marks on histones. Functional diversity in these complexes depends on the type of RCOR included, which modulates the catalytic activity of the complex. Here, we studied the duplicative history of the *RCOR* and *LSD* gene families and analyzed the evolution of their interaction. We found that *RCOR* genes are the product of the two rounds of whole-genome duplications that occurred early in vertebrate evolution. In contrast, the origin of the *LSD* genes traces back before to the divergence of animals and plants. Using bioinformatics tools, we show that the RCOR and LSD1 interaction precedes the *RCOR* repertoire expansion that occurred in the last common ancestor of jawed vertebrates. Overall, we trace LSD1-RCOR complex evolution and propose that animal non-model species offer advantages in addressing questions about the molecular biology of this epigenetic complex.

[1] Department of Cellular and Molecular Biology, Faculty of Biological Sciences, Pontificia Universidad Católica de Chile, Santiago, Chile. [2] Departamento de Ciencias Biomédica, Facultad de Medicina, Universidad Católica del Norte, Coquimbo, Chile. [3] Department of Biological Sciences, Columbia University, New York, NY, USA. [4] Department of Genetics and Development, Columbia University Medical Center, New York, NY, USA. [5] Departamento de Ciencias Farmacéuticas, Facultad de Ciencias, Universidad Católica del Norte, Antofagasta, Chile. [6] Facultad de Medicina y Ciencia, Universidad San Sebastián, Valdivia, Chile. [7] Integrative Biology Group, Valdivia, Chile. [8] Millennium Nucleus of Ion Channel-Associated Diseases (MiNICAD), Valdivia, Chile. [9] These authors contributed equally: Montserrat Olivares-Costa, Gianluca Merello Oyarzún, Daniel Verbel-Vergara. ✉email: mandres@bio.puc.cl; juan.opazo@uss.cl

Lysine-specific demethylase 1A (LSD1, KDM1A, BHC110, AOF2) is an epigenetic enzyme that represses gene expression by erasing transcriptionally permissive histone modifications[1,2]. LSD1 function and stability depend on forming a stable complex with a member of the RCOR family of transcriptional corepressors[3–5]. Mammalian LSD1 has a central SWIRM (Swi3p, Rsc8p, and Moira) domain and a C-terminus amino oxidase (AOD) catalytic domain (Fig. 1). The AOD domain is interrupted by an alpha-helical tower domain of 92 amino acids that allows the interaction of LSD1 with the RCOR (REST corepressor) family of proteins, composed in mammals by RCOR1 (CoREST, CoREST1), RCOR2, and RCOR3[3,6,7]. RCOR proteins share a characteristic structure, including three functional domains (Fig. 1). An ELM2 (homology 2 of Egl-27 and MTA1) domain, followed immediately by a SANT (Swi3, Ada2, N-Cor, and TFIIIB) domain. A second SANT domain localized at the C-terminus of RCOR proteins allows the interaction with nucleosomal DNA[8]. The linker domain, sufficient for binding LSD1[4], hugs the tower domain[7,9].

Although the three RCORs interact with LSD1, the catalytic properties of the different complexes differ and functionally have been associated with different biological processes. For example, RCOR1 regulates differentiation into various cell lineages[10–12] and represses the expression of viral genomes[13], while RCOR2 maintains pluripotency and proliferation of embryonic stem cells[14]. Although both have relevant roles in central nervous system development[15,16], RCOR3 has received less attention, with one report showing RCOR3-mediated competitive inhibition of LSD1-dependent histone demethylation[17].

Regarding the evolution of these gene families, not much is known, although studies have reported distinct evolutionary patterns in plant and animal LSD (LSD1 and LSD2) proteins[18]. Today, the availability of whole-genome sequences in a wide range of taxonomic groups opens an outstanding opportunity to shed light on the evolution of gene families. Understanding the duplicative history of gene families is required, among other things, to make biologically meaningful comparisons. Although several questions can be asked by analyzing the gene repertoire in representative species of a given taxonomic group, some questions related to the LSDs and RCORs require special attention. For example, when did the expansion of the RCOR repertoire happen? Are they the product of whole-genome duplications or loci-specific duplications? What is the conservation pattern of the different domains of the LSD and RCOR genes?

In this work, we studied the evolutionary history of the LSD and RCOR gene families in animals. Our results suggest that RCOR genes are ohnologs and that their diversification occurred in the last common ancestor of jawed vertebrates. At the same time, the origin of LSD paralogs is much more ancient. According to their phyletic distribution, LSD1 and RCOR are widespread in metazoans. Our structural analyses suggest that the RCOR and LSD1 proteins present in the last common ancestor of jawed vertebrates are able to interact, suggesting that the LSD1-RCOR interaction precedes the diversification of the RCOR genes.

## Results and discussion

**The RCOR gene repertoire expanded in the ancestor of jawed vertebrates.** To understand the duplicative history of the RCOR genes, we reconstructed gene phylogenies with different taxonomic samplings. The first analysis aimed to understand the evolution of RCOR genes in vertebrates (Fig. 2), whereas in the second, our sampling effort included representative species of all main groups of animals (Fig. 3).

In the first analysis, our maximum-likelihood tree recovered well supported clades corresponding to RCOR1 sequences from vertebrates and RCOR2, and RCOR3 sequences of jawed vertebrates (i.e., gnathostomes) (Fig. 2). In this tree, the clade containing RCOR1 sequences was recovered sister to RCOR3 clade; however, this relationship is not supported (Fig. 2). The clade containing RCOR2 sequences from jawed vertebrates was recovered sister to the RCOR1/RCOR3 clade (Fig. 2). Our analysis recovered a clade containing RCOR sequences from two jawless vertebrates (i.e., cyclostomes) species (inshore hagfish and sea lamprey) sister to the RCOR1 clade from jawed vertebrates (Fig. 2). This topology suggests different evolutionary scenarios: (1) the RCOR genes diversified in the last common ancestor of vertebrates, and jawless vertebrates only retained one copy (RCOR1) or (2) the RCOR genes diversified in the last common ancestor of jawed vertebrates and the sister group relationship recovered in our gene tree is a phylogenetic artifact. This last scenario could be possible, given that resolving orthology between jawless and jawed vertebrates is a complex evolutionary problem because of the compositional biases of the former group[19,20]. In addition, resolving phylogenetic relationships among vertebrates needs a taxonomic sampling that includes more than just vertebrate species.

To further understand if the RCOR genes diversified in the ancestor of vertebrates or jawed vertebrates, we performed a phylogenetic analysis extending our sampling to representative species of all main groups of animals and reducing the representation of vertebrates. In addition to showing the presence of a single copy gene in all major groups of animals other than vertebrates, our phylogenetic tree resolves the sister group relationship between jawless and jawed vertebrates (Fig. 3), and it is consistent with our second proposed scenario (Fig. 3). We recovered the monophyly of the vertebrate clade containing RCOR sequences with strong support (100/1/100, Fig. 3). The clade containing RCOR sequences from jawless vertebrates was recovered sister to the group containing the RCOR1, RCOR2, and RCOR3 clades from jawed vertebrates (Fig. 3), suggesting that the diversification of the RCOR genes occurred between 615 and 473 million years ago[21] in the last common ancestor of jawed

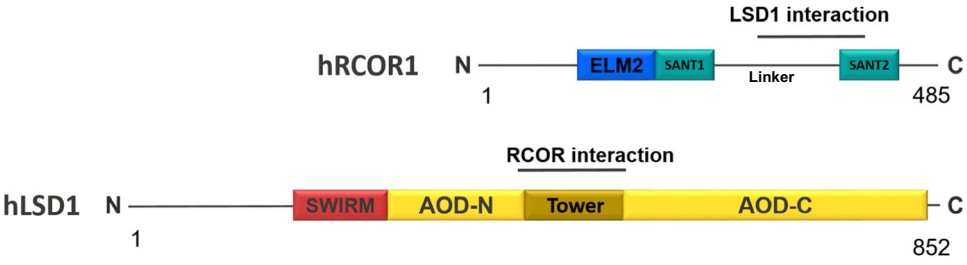

**Fig. 1 Schematic representation of human RCOR1 and LSD1 proteins highlighting their functional domains.** ELM2 (Egl-27 and MTA1 homology 2) domain; SANT1 (Swi3, Ada2, NCoR, and TFIIIB) and SANT2 domains; SWIRM (Swi3, Rsc8, and Moira) domain; AOD-N (N-terminus amino oxidase domain); AOD-C (C-terminus amino oxidase domain).

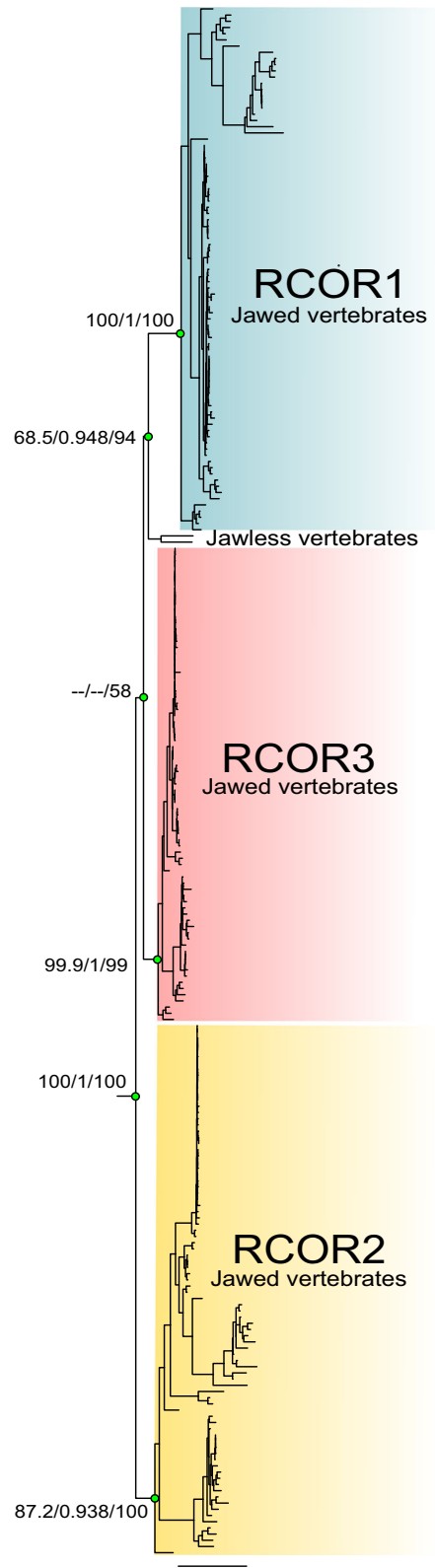

**Fig. 2 Maximum-likelihood tree showing relationships among *RCOR* genes of vertebrates.** Numbers on the nodes correspond to support from the Shimodaira–Hasegawa approximate likelihood-ratio test, approximate Bayes test, and ultrafast bootstrap values. The scale denotes substitutions per site and colors represent gene lineages. Mitotic deacetylase-associated SANT domain protein (MIDEAS) sequences from human (*Homo sapiens*), gorilla (*Gorilla gorilla*), mouse (*Mus musculus*), and ferret (*Mustela putorius furo*) were used as outgroup (not shown). For more details of the species included in the analysis, refer to Supplementary Data 1.

Although the most accepted hypothesis invokes two rounds of WGD during the evolutionary history of vertebrates, the timing of these duplication events is still a matter of debate[27,28]. The most recent hypothesis suggests that one of the duplications occurred before the divergence of cyclostomes and gnathostomes, while the second took place in the gnathostome ancestor[27,28]. This scenario suggests that the vertebrate ancestor had two *RCOR* genes, a gene repertoire inherited by cyclostomes and gnathostomes. One of the copies was lost in the first group, while in the ancestor of gnathostomes, the second round of whole-genome duplication produced four *RCOR* copies, and subsequently, one copy was lost. Genes that originated as results of WGDs are called ohnologs, in honor of Susumu Ohno, who was the first to propose the occurrence of two rounds of WGDs early in the evolution of vertebrates[29]. The expansion of *RCOR* genes in jawed vertebrates suggests that they could be the result of the WGDs occurred during the evolution of vertebrates. After checking the repository of genes retained from WGDs in the vertebrate genomes[30], we confirmed that the *RCOR* genes are indeed the product of the vertebrate-specific WGDs. Based on this evidence, we propose ohnologs as the appropriate term to describe their homology relationship.

The role of each of the RCOR proteins is not yet clear. RCOR1 plays a specific role in keeping viral genomes in latency in neurons[31]. Studies in RCOR1 null mice reveal a crucial role of this protein in erythropoiesis and the proliferation of regulatory T cells[11,32]. On the other hand, RCOR2 is expressed in embryonic stem cells, regulating their proliferation and pluripotency[14]. In the case of RCOR3, an isoform without the SANT2 domain has been shown to play antagonistic roles compared to RCOR1 and RCOR2 during myeloid cell lineage differentiation[17] Together with other reports, this evidence highlights the contribution of WGDs to functional diversification among *RCOR* ohnologs, supporting the pivotal role of WGDs in the origin of biological novelties.

***LSD1* and *LSD2* were present in the ancestor of all animals**. Given the specificity of the interaction of RCOR proteins exclusively with LSD1 but not with its paralog Lysine-specific demethylase 1B (LSD2, KDM1B), we sought to investigate the evolutionary history of the *LSD* gene family in animals. To accomplish this, we performed two phylogenetic analyses. In the first, we included representative species of the main groups of vertebrates (Fig. 4), whereas in the second, our sampling effort expanded to the main groups of animals (Supplementary Fig. 1).

In our first maximum-likelihood tree, we recovered the monophyly of *LSD1* and *LSD2* genes from vertebrates (Fig. 4), suggesting that the ancestor of the group, which existed between 676 and 615 million years ago[21], had both paralogs. To gain a deeper understanding of the origin of these two genes, we analyzed their duplicative history by including representative species of all main groups of animals. Our phylogenetic analyses also recovered the monophyly of each paralog, *LSD1* and *LSD2*, suggesting that the genome of the ancestor of all animals had both paralogs (Supplementary Fig. 1). Thus, our findings are consistent

vertebrates, after the divergence from jawless vertebrates. Among jawed vertebrates, our gene tree recovered the sister group relationship between RCOR1 and RCOR2, while the RCOR3 clade was recovered sister to the RCOR1/RCOR2 clade (Fig. 3).

It is widely accepted that the evolution of vertebrates was shaped by ancient whole-genome duplications (WGDs)[22–26].

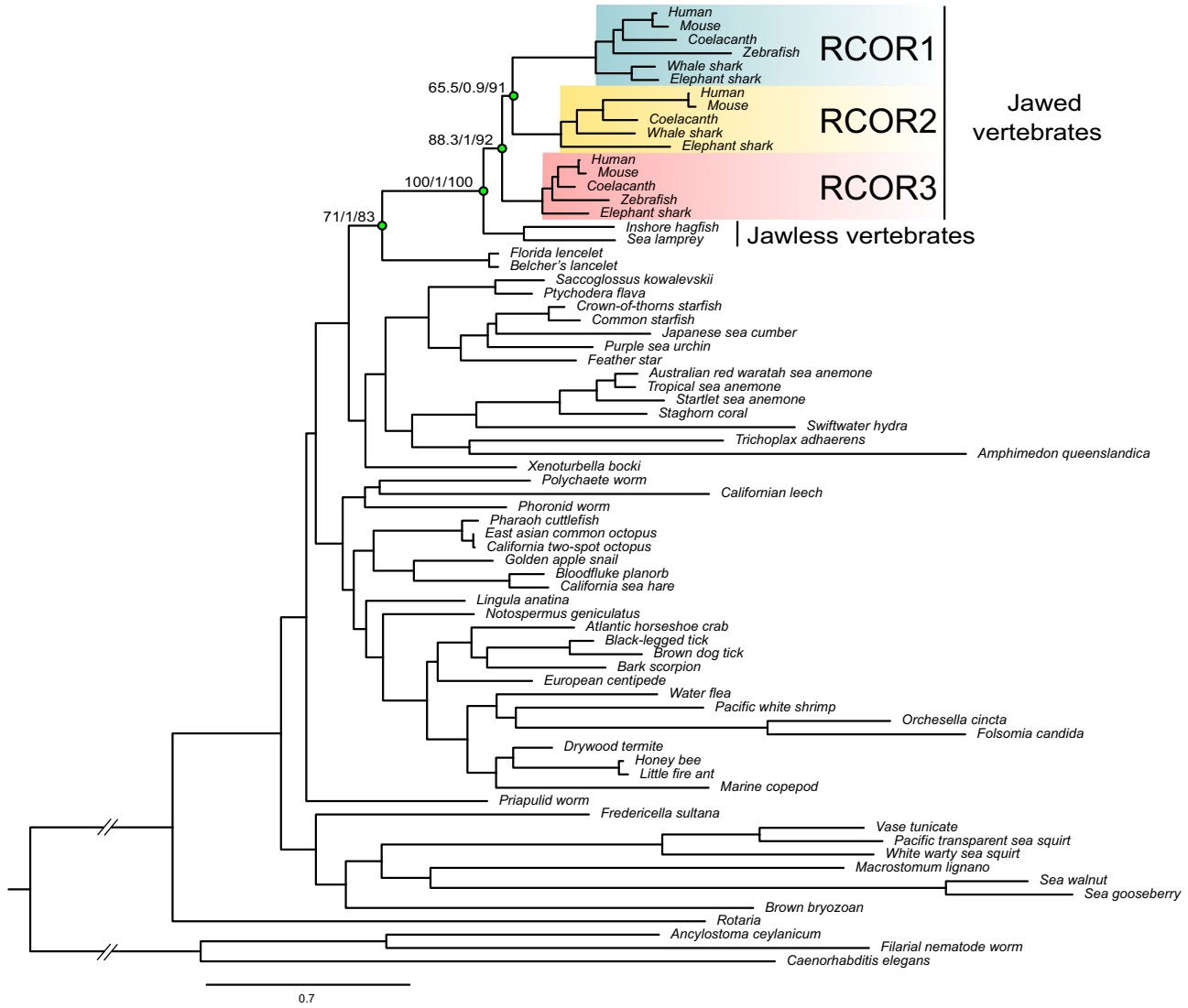

**Fig. 3 Maximum-likelihood tree showing relationships among *RCOR* genes of metazoa.** Numbers on the nodes correspond to support from the Shimodaira–Hasegawa approximate likelihood-ratio test, approximate Bayes test, and ultrafast bootstrap values. The scale denotes substitutions per site and colors represent gene lineages. Mitotic deacetylase-associated SANT domain protein (MIDEAS) sequences from human (*Homo sapiens*), gorilla (*Gorilla gorilla*), mouse (*Mus musculus*), and ferret (*Mustela putorius furo*) were used as outgroup (not shown). For more details on the species included in the analysis, refer to Supplementary Data 2.

with previous reports suggesting an early origin of *LSD* genes before the divergence of animals and plants[18].

**Phyletic distribution of *RCOR* and *LSD* genes**. To better describe the evolutionary history of *RCOR* and *LSD* genes in animals, we analyzed their phyletic distribution, i.e., their presence and absence in different animal groups. RCOR1, RCOR2, and RCOR3 were found in all major groups of vertebrates other than cyclostomes, which possess a single gene copy (named *RCOR1/2/3*) (Figs. 2, 3, and 5a and Supplementary Data 1). Interestingly, although RCOR1 and RCOR3 are present in all examined bird species, RCOR2 is restricted to species belonging to the orders Psittaciformes, Passeriformes, Accipitriformes, and Anseriformes (Supplementary Data 1). A comparison of the chromosomal regions (flanked by NAA40 and MARK2 genes) in chicken and painted turtle suggests the absence of the *RCOR2* gene in the chicken genome (Galliformes) (Fig. 5b). By contrast, in the New Caledonian crow (Passeriformes), conserved regions spanned all 12 exons of the *RCOR2* gene (Fig. 5b). Given the

phyletic distribution of RCOR2 in different orders of birds (Supplementary Data 1), we suggest that the *RCOR2* gene was lost independently in different bird lineages.

RCOR1 and RCOR2 play preponderant roles in the development of the cerebral cortex in mammals. Mice null for RCOR2 in neural lineage cells show decreased neocortex thickness and brain size[15], whereas knocking down *RCOR1* in later stages of development alters the differentiation and migration of cortical neurons[33]. Although there are no studies focused on the functional roles of RCORs in birds, it would be interesting to compare the formation of the dorsal telencephalon between birds with the full complement of *RCOR* genes with those lacking RCOR2.

In invertebrates, we found *RCOR* genes in 19 of 35 phyla analyzed. Specifically, RCOR was found in all analyzed phyla but in Tardigrada, Kinorhyncha, Loricifera, Nematomorpha, Sipuncula, Orthonectida, Micrognathozoa, Rhombozoa, Onychophora, Gnathostomulida, Gastrotricha, Entoprocta, Cycliophora, Chaentognatha and Acanthocephala. (Supplementary Data 2). *LSD1* and *LSD2* genes were found in all main vertebrate groups (Fig. 4

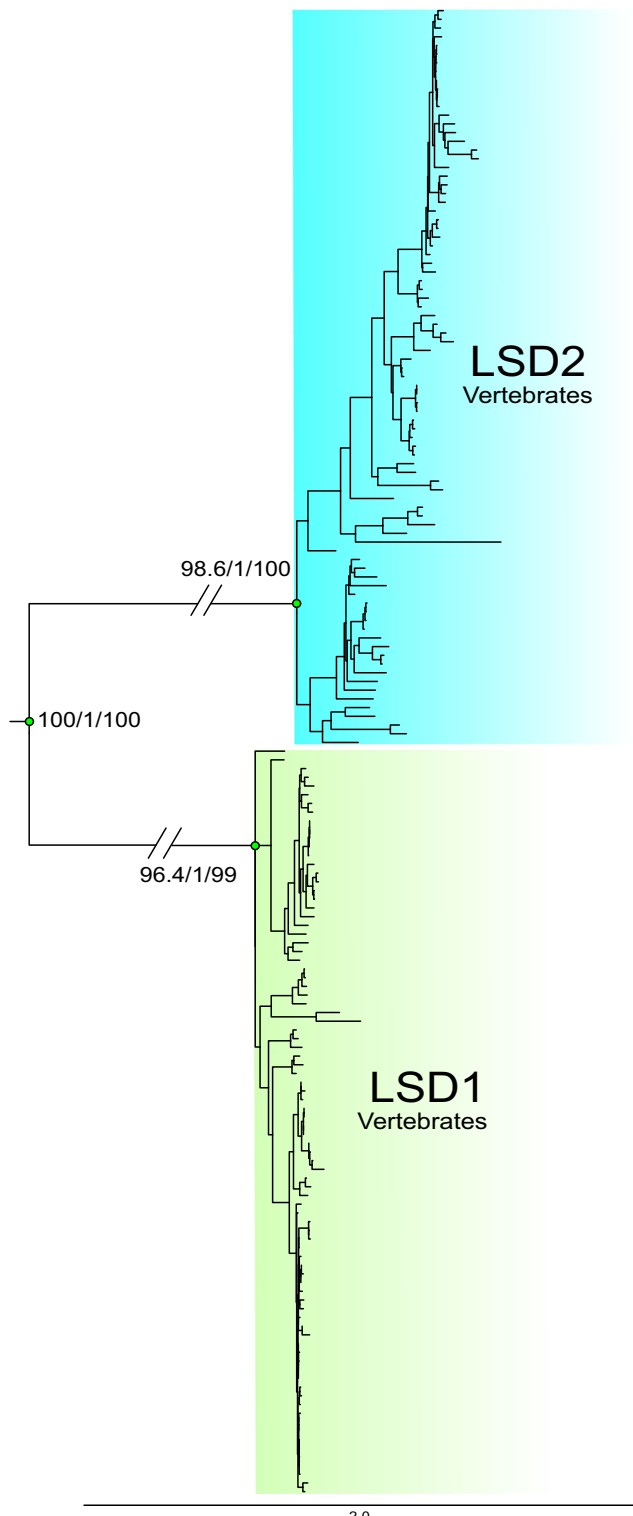

**Fig. 4 Maximum-likelihood tree showing relationships among *LSD* genes of vertebrates.** Numbers on the nodes correspond to support from the Shimodaira–Hasegawa approximate likelihood-ratio test, approximate Bayes test, and ultrafast bootstrap values. The scale denotes substitutions per site and colors represent gene lineages. Monoamine-oxidases (*MAO-A* and *MAO-B*) sequences from human (*Homo sapiens*), chicken (*Gallus gallus*), spotted gar (*Lepisosteus oculatus*), and coelacanth (*Latimeria chalumnae*) were used as outgroup (not shown). For more details on the species included in the analysis, refer to Supplementary Data 3.

and Supplementary Data 3). L*SD1* was found in 22 of 34 phyla studied, excluding the invertebrates phyla Kinorhyncha, Loricifera, Nematomorpha, Sipuncula, Orthonectida, Micrognathozoa, Onychophora, Gnathostomulida, Gastrotricha, Entoprocta, Cycliophora, and Chaetognatha. On the other hand, *LSD2* was not found in some Arthropods, as previously reported[18], and Placozoa, Tardigrada, Kinorhyncha, Loricifera, Nematomorpha, Sipuncula, Rotifera, Platyhelminthes, Orthonectida, Micrognathozoa, Rhombozoa, Onychophora, Gnathostomulida, Gastrotricha, Entoprocta, Cycliophora, Chaetognatha, and Acanthocephala (Supplementary Data 4 and Supplementary Fig. 2).

The results of our analysis indicate that there is a greater abundance of species in which *RCOR* genes co-exist with *LSD1*, as compared to species in which only one of these genes is present (Supplementary Data 2 and 4). Nevertheless, there exist certain instances that deviate from the norm. For example, the organism known as the water bear (*Ramazzottius varieornatus*) exhibits the presence of *LSD1* but not *RCOR*. This example would imply that LSD1 is functionally independent of RCOR in some animal lineages. This species and others could provide a suitable platform to investigate RCOR (or LSD1) functions that are LSD1 (or RCOR)-independent to further understand the diverse range of biological processes in which these proteins are involved.

**Mammals and turtles possess the regulatory sequences to express the *LSD1-8a* specifically in neurons.** The *LSD1* gene possesses nineteen exons in humans, yet alternative splicing can incorporate two additional exons, generating four splice variants. Exon 2a (E2a) (60 bp) encodes for 20 amino acids, and microexon 8a (E8a) (12 bp) encodes for four amino acids. Isoforms, including microexon E8a (*LSD1-8a* and *LSD1 2a-8a*), are exclusively expressed in neurons and are called neuronal LSD1 (neuroLSD1)[34]. E8a microexon inclusion occurs during neuronal differentiation and enhances neurite morphogenesis in mammals[34]. Further research has demonstrated a role for neuroLSD1 in modulating behavior[15,35]. NeuroLSD1-KO mice display less anxiety in various behavioral tests[35], a diminished response to epileptogenic stimuli[35], and defects in spatial learning and memory[36,37]. When discovered, neuroLSD1 was described as a mammalian-specific protein[34], although the same microexon was later identified in turtles[36], and an 8a-like exon was also described in zebrafish[38].

Due to E8a's relevance in neuronal development and behavior, and the lack of a systematic study of the E8a phyletic distribution, we analyzed the presence of the E8a sequence at the *LSD1* gene in representative species of all main groups of vertebrates. To annotate E8a sequences, we selected the intronic region between exons 8 and 9, using the human *LSD1* genomic sequence as reference. We considered as E8a, sequences aligning with human (*Homo sapiens*) E8a[34] or zebrafish (*Danio rerio*) E8a-like[38], including conserved splice donor and acceptor sites (GT and AG, respectively). Our analyses confirmed that the E8a microexon (i.e., DTVK) is also present in turtles (Fig. 5a), consistent with previous reports[34,36]. On the other hand, crocodiles, amphibians, lizards, and snakes do not have the E8a sequence (Fig. 5a). Interestingly, the intronic region of these species is shorter than those in mammals and turtles (mean of 1537.5 bp in crocodiles, amphibians, lizards, and snakes, whereas 9587.2 bp in mammals and turtles (Supplementary Data 5)). In birds, we found the E8a sequence exclusively in species belonging to the orders of Phaethontiformes, Gaviiformes, Opisthocomiformes, and Accipitriformes (Fig. 5a and Supplementary Data 5). In addition, we found a six amino acid coding sequence aligning with the E8a-like exon in bony fish, cartilaginous fish, and cyclostomes, (Supplementary Data 5)[38]. Thus, the microexon E8a appears to have

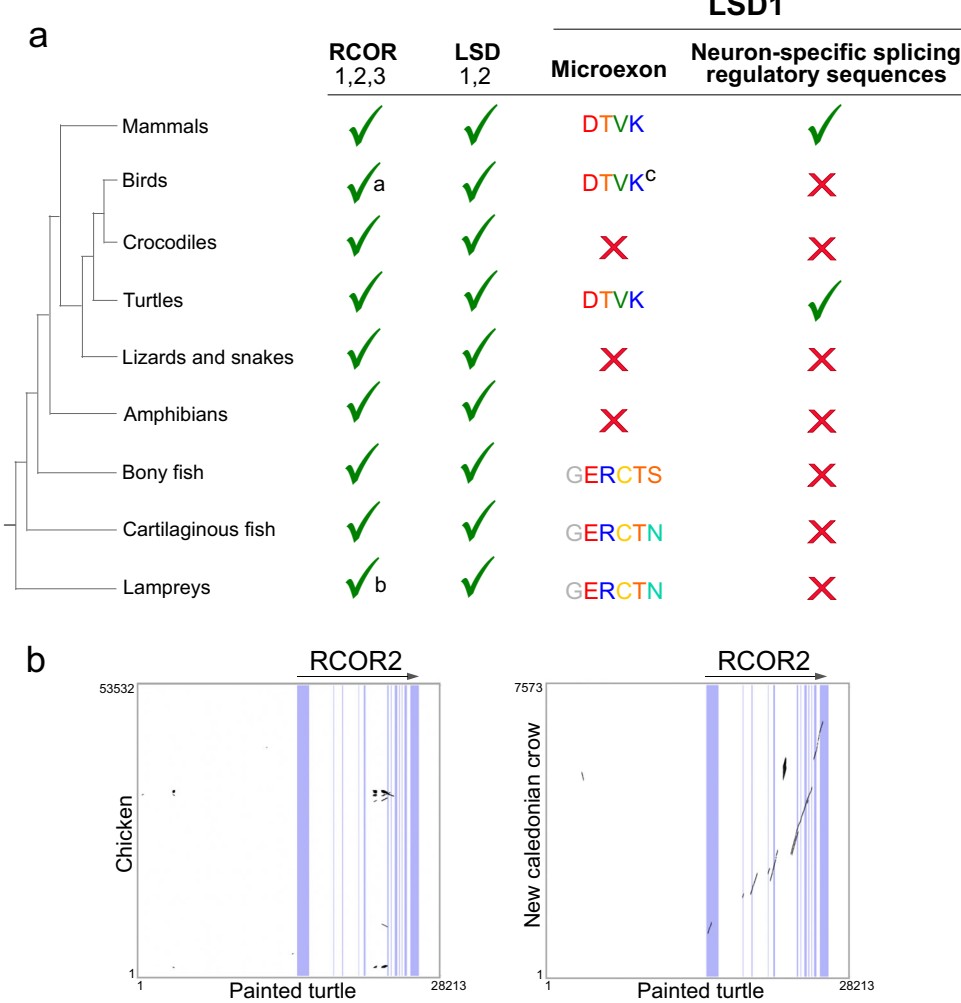

**Fig. 5 Phyletic distributions of *RCOR* and *LSD* genes, *LSD1* microexon, and neuron-specific splicing regulatory sequences in vertebrates. a** Distribution of the *RCOR* and *LSD* genes, *LSD1* microexon, and neuron-specific splicing regulatory sequences in main groups of vertebrates. **b** Dot-plot of pairwise sequence similarity between the *RCOR2* gene of the painted turtle (*Chrysemys picta*) and the corresponding syntenic region in the chicken (*Gallus gallus*) and New Caledonian crow (*Corvus moneduloides*). [a]*RCOR2* is present only in the bird orders Psittaciformes, Passeriformes, Accipitriformes, and Anseriformes. [b]Lampreys have a single copy of the *RCOR* gene. [c]Some species of birds have a -DTVE- microexon. For more details on the species included in the analysis, refer to Supplementary Data 5.

been present in the vertebrate ancestor and lost multiple times as well as diverged (e.g., tablemammal vs. fish sequence) during the radiation of the group.

The E8a retention into *LSD1* transcripts is regulated by three splicing factors (NOVA1, FUBP, and SRRM4). A palindromic sequence located ~300 bp toward the 3′ of E8a can trap the exon and its donor and acceptor splicing sites into a double-stranded RNA structure[39,40]. SRRM4 binds the UGCUGC motif upstream of the splice acceptor site of the exon E8a[39], and together with NOVA1 and FUBP, they form a complex that can maintain a single-stranded pre-mRNA and therefore elicit exon E8a retention. We were not able to find the palindromic sequence and the UGCUGC motif (from now, splicing regulatory sequences (SRS)) in most vertebrates other than mammals and turtles (Fig. 5a). Given the presence of E8a or E8a-like and the absence of SRS in most vertebrate groups, we hypothesize that the expression of *LSD1-8a* is not restricted to neuronal tissue in these animals. This finding agrees with previous results in zebrafish in which *LSD1-8a*-like is ubiquitously expressed[38]. On the other hand, *LSD1-8a* might be neuron-specific in turtles, although direct evidence for that is still lacking. If that is the case, though, turtles could serve

as an interesting model to study the role of *LSD1-8a* in neuron development and plasticity.

**RCOR ohnologs linker and LSD1 tower domains display lower evolutionary conservation in comparison to their other functional domains**. LSD1 and RCOR have characteristic functional domains (Fig. 1). The SWIRM and catalytic domains (AOD) in LSD1 and the interaction domains ELM2, SANT1 (for HDAC1/2), and SANT2 (for nucleosomal DNA and histone octamer) in RCORs. In addition, LSD1-RCOR binding occurs between LSD1's tower domain and RCOR's linker region. Although all RCORs bind with similar affinity to LSD1, the resulting complexes differ in protein composition and catalytic capacities[6,8,17], suggesting that expansion of the *RCOR* repertoire as a product of the whole-genome duplications that occurred early in the evolution of vertebrates have extended the functional capabilities of the LSD1-RCOR complexes.

To delve into the evolution of the RCOR and LSD1 molecular interaction, we studied the conservation of their functional domains in jawed vertebrates. To this end, multiple sequence

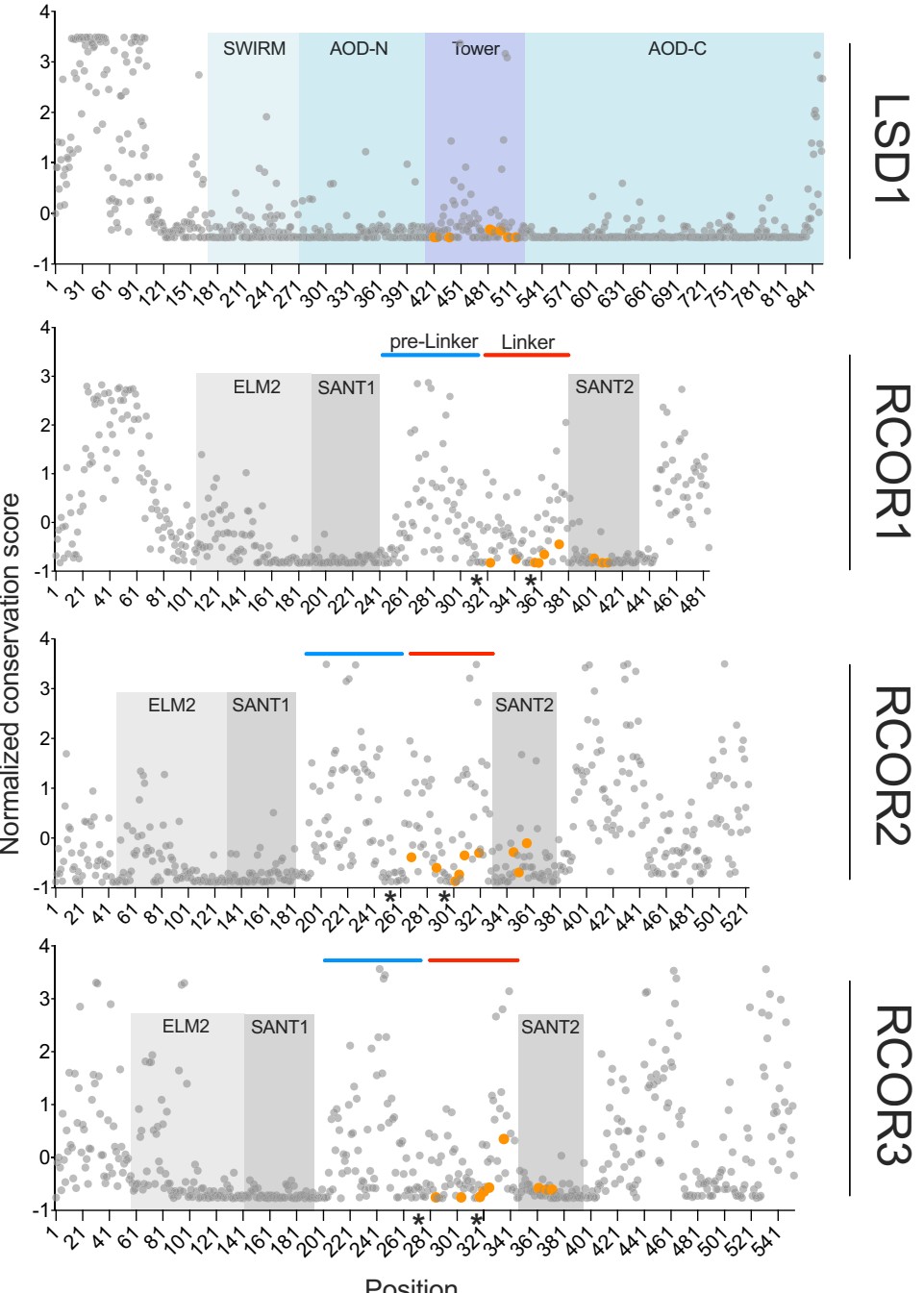

**Fig. 6 LSD1 and RCORs normalized conservation scores of jawed vertebrates.** Orange dots represent the conservation scores of the key amino acids for LSD1-RCOR interaction[49]. The red line in RCORs denotes the linker domain required for interaction with LSD1. The blue line shows a pre-linker highly variable sequence with intrinsically disordered characteristics. Black asterisks indicate conserved sequence patches. Position numbers correspond to human proteins.

alignments for each protein, incorporating representative species of all major groups of jawed vertebrates (mammals, reptiles, birds, amphibians, bony fish, and cartilaginous fish), were constructed. These alignments were used as an input for the ConSurf web server[41–43] to calculate positional conservation scores (Fig. 6). ConSurf outputs either continuous conservation scores, which were used to construct Fig. 6 or creates discrete groups, ranked 1 (variable) − 9 (conserved) which were used to construct Fig. 7. When comparing each domain's conservation scores in the case of RCOR1, we found that its linker domain is significantly less conserved (higher normalized core) than SANT1 ($p < 0.0001$) and

SANT2 ($p < 0.0001$) domains. Analyses for RCOR2 rendered similar results with its linker domain significantly less conserved than SANT1 ($p < 0.0001$), SANT2 ($p = 0.0016$) and ELM2 ($p < 0.0001$). Finally, RCOR3s' linker domain was also found to be less conserved than SANT1 ($p < 0.0001$) and SANT2 ($p = 0.0003$) (Fig. 6 and Supplementary Fig. 4).

A more detailed examination of the conservation patterns inside the linker domain revealed two highly conserved patches of 11–28 amino acids separated by a less conserved segment, which are particularly visible in RCOR2. Both patches are also present in RCOR1 and RCOR3 (Fig. 6, black asterisks). Interestingly, in

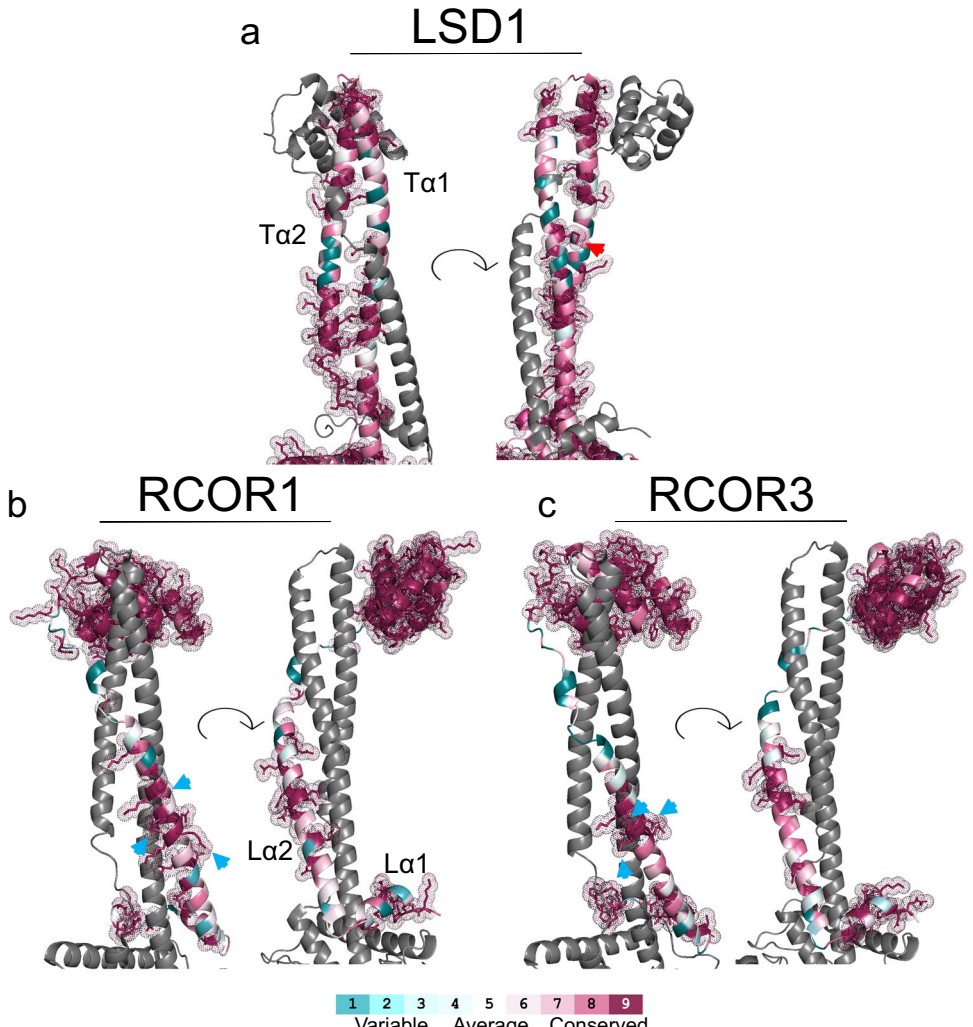

**Fig. 7 Color-coded 3D amino acid conservation of LSD1-RCOR protein complexes. a** LSD1 tower domain residue conservation projected onto the structure of human LSD1-RCOR1 (PDB: 2V1D). The red arrow shows Lys447 residue (see text). **b** RCOR1 linker-SANT2 domain residue conservation projected onto the structure of human LSD1-RCOR1 (PDB: 2V1D). Light blue arrows show Arg347, Gln350, and Gln354 residues (see text). **c** RCOR3 linker-SANT2 domain residue conservation projected onto the structure of human LSD1-RCOR3 (PDB: 4CZZ). Light blue arrows show Gln315, Asn316, and Gln319 residues (see text).

RCOR2, the first patch consisted of 18 residues located immediately before the Lα1 helix of the linker domain (Supplementary Fig. 5a, red color). In RCOR1 it included 12 amino acids and in RCOR3 18 amino acids before the Lα1 helix. In both RCOR1 and RCOR3, it also incorporated the first amino acids of the Lα1 helix (Supplementary Fig. 5b, c, red color). The second patch was located in the middle section of the Lα2 helix in all three proteins, consistent with the key role of the Lα2 helix in LSD1 binding. These findings show that in all three RCOR proteins, a short segment that immediately precedes the Lαα1 helix, with unclear structure and little examination to date, has been under selective pressure during jawed vertebrate evolution, suggesting functional relevance.

Human RCOR1/2/3 linker domains are preceded by a region that has not been crystallized to date. Interestingly, this region (from now on pre-linker) appears as a marked increase in variability after the SANT1 domain (Fig. 6, blue line). While performing predictions of functional domains with InterPro[44], we found that the pre-linker constitutes an intrinsically disordered region (IDR) present in human RCOR1, RCOR2, and RCOR3.

We confirmed this finding using three other IDR predictors: IUPred3[45], flDPnn[46] and NetSurfP − 3.0[47]. We next wondered if, notwithstanding the low conservation of the pre-linker region (Fig. 6, blue line), the intrinsically disordered characteristic would be maintained in different jawed vertebrates. To address this, we used the same disorder predictors to analyze RCORs from representative species of the main groups of jawed vertebrates. We found that the pre-linker segment constitutes an IDR in all species analyzed (Human, Chinese softshell turtle, Zebra finch, Small tree finch, New Caledonian crow, Western clawed frog, Amazon Molly, Thorny skate and Whale shark) in all three RCORs (Supplementary Fig. 6, between vertical red lines). This suggests that in this linker-neighboring region of the *RCOR* ohnologs, selective pressure is exerted on disorder capabilities rather than on sequence conservation, which agrees with the importance of IDRs in the functioning of chromatin-associated proteins[48].

Analysis of LSD1s' functional domains revealed a similar situation to that of RCORs. The RCOR-interacting tower domain showed significantly lower conservation than the other

functional domains of the protein, namely the SWIRM, amine oxidase N-terminal ($p = 0.0007$) and amine oxidase C-terminal domains ($p < 0.0001$) between which it is inserted (Fig. 6 and Supplementary Fig. 4).

Lower conservation of the linker domain in RCORs and tower domain in LSD1 was surprising given their role in forming the molecular complex. We analyzed the conservation of amino acids key to the interaction to get insight into this intriguing aspect. To this end, we conducted a comparative analysis between RCOR1 and RCOR3 based on their 3D structures in complex with LSD1[6,7,49]. Using this information, we searched for the conservation of those key amino acids in equivalent positions in RCOR2. Interaction interfaces of LSD1-RCOR1 and LSD1-RCOR3 complexes were divided into four segments, I to IV (Supplementary Fig. 3). In segment I, two salt bridges are formed between two Asp residues of RCOR1 and two Lys residues of LSD1. These residues are in the same position in RCOR3 forming equivalent salt bridges with LSD1. Segment II contains two salt bridges and one hydrogen bond between RCOR1 and LSD1. These are also conserved and in close contact in the LSD1-RCOR3 complex. In segment III, a Lys in position 371 in RCOR3 substitutes the Arg in RCOR1 for the salt bridge with Asp495 of LSD1. Section IV is not essential for LSD1-RCOR1 binding[4] but might aid LSD1s' indirect interaction with the DNA or the histone octamer[50]. In segment IV, Lys397, Asp401, and Asp407 are adequately located to form two additional salt bridges between RCOR1-LSD1 and RCOR3-LSD1 (Supplementary Fig. 3).

The residues mentioned above are either conserved or substituted by chemically equivalent amino acids in RCOR2, except for Asp320, which is replaced with a Gly (Supplementary Fig. 3). High conservation of key amino acids in relevant positions for LSD1 interaction in the RCOR proteins (highlighted in orange in Fig. 6) indicates that they evolved before the repertoire expansion and have mainly remained conserved for the last 615 million years.

Lower mean conservation scores in the linker and tower domains could indicate a lower selective pressure on these regions relative to the other functional domains. Alternatively, linker and Tower domain variability could have made possible the flexible encounter of different RCORs' with LSD1 and the modulation of the catalytic activity that these exert on LSD1.

**Residues in the most conserved positions in the primary structure of RCOR1 and RCOR3 linker domains constitute the interaction interface with LSD1.** Next, we expanded our analysis to all residues that comprise the tower domain of LSD1 and linker regions of RCORs (i.e., not only key residues for interaction). We investigated whether there is a relationship between the conservation score of a particular residue and its position in the 3D structure of the complex. Again, these analyses were restricted to RCOR1 and RCOR3, given their known 3D structure in complex with LSD1.

In the case of LSD1, 41.0% of the residues of the tower domain were found to be highly conserved, which is significantly higher than expected by chance ($X^2$ (df = 1, $N = 105$) = 90.4, $p < 0.0001$) (Fig. 7a, dark red). These residues are enriched in the interface between LSD1 and RCOR1 as 61.9% of interface residues are highly conserved. These amino acids are distributed along the four segments analyzed in Supplementary Fig. 3. The other group, i.e., the highly conserved amino acids that do not participate in the interaction interface, are located mostly on the basal/middle section of the Tα2 helix and the distal segment of the Tα1 helix (Fig. 7a). The specific function of these amino acids remains to be explored.

As for RCOR1, 33.7% of residues in its linker region displayed highly conserved scores, this is significantly more than expectations by chance ($X^2$ (df = 1, $N = 74$) = 32.4, $p < 0.0001$) (Fig. 7b). We observed enrichment of this group of amino acids in the contacts between RCOR1 and LSD1, with 59.1% of them classified as highly conserved. The majority of these residues are located in the Lα2 helix. Interestingly, however, three highly conserved residues in the Lα2 helix, Arg347, Gln350, and Gln354 (Fig. 7b, light blue arrows) are oriented toward the opposite direction of LSD1. The function of these amino acids is currently unknown, but given their high conservation, they deserve further attention. Finally, in the case of RCOR3, 31.6% of the residues in its linker domain are highly conserved, which is higher than expected by chance ($X^2$ (df = 1, $N = 76$) = 28.4, $p < 0.0001$). In addition, 35% of the residues in the RCOR3-LSD1 interface are highly conserved (Fig. 7c). Furthermore, for RCOR3, highly conserved amino acids are not solely enriched in the Lα2 helix, but are more evenly distributed between the Lα1 and Lα2 helices. In addition, and as seen with RCOR1, three Lα2 residues, Gln315, Asn316, and Gln319, oriented themselves on the opposite side of the Lα2 helix, away from the interaction interface (Fig. 7c, light blue arrows).

Thus, this analysis sheds light on several conserved segments that we hypothesize fulfill relevant molecular functions in each protein. As an example, Lys447 (Fig. 7a, red arrow) at LSD1s' tower domain, a highly conserved residue that does not participate in the interaction interface, has been previously shown to inform the structural relationships between LSD1 and HDAC1 within the RCOR1 ternary core complex[51]. Also, the most conserved amino acids in RCOR1 and RCOR3 linker domains were predictive of LSD1 interaction capabilities. Most interesting are the residues that proved highly conserved but whose molecular function has yet to be explored. Specifically, the outward-facing arginine, glutamine, and asparagine of RCOR1 and RCOR3 hold promise for further analyses.

**Evidence of RCOR and LSD1 interaction preceding *RCOR* repertoire expansion in jawed vertebrate ancestors based on structural analyses.** Given the evolutionary pattern of repertoire expansion of *RCOR* genes in the jawed vertebrate ancestor and the lack of a systematic assessment of RCOR-LSD1 interaction in non-vertebrate species, we wondered if the single RCOR protein present in the jawed vertebrate ancestor before the *RCOR* repertoire expansion was able to interact with the corresponding LSD1 ancestral protein. To investigate this matter, we manually curated RCOR1, RCOR2, RCOR3, and LSD1 sequences to reconstruct the RCOR and LSD1 proteins present in the ancestor of jawed vertebrates (Fig. 8a, b and Supplementary Data 6).

We then predicted the resulting structure of the ancestral protein complex using ColabFold[52] and AlphaFold-Multimer[53]. The resulting tower-linker interaction structure showed predicted local-distance difference test (pLDDT) values >90 (reflecting high model confidence) in all regions of RCOR and LSD1 except for the connection between the Lα2 helix of the linker domain and the SANT2 domain in RCOR (Supplementary Fig. 7a, b). This short segment was therefore omitted from further analyses. In addition, the inter-domain accuracy of the tower-linker domains in the heterodimer was confidently predicted as seen by the predicted aligned error (PAE) (Supplementary Fig. 7c).

Overall, the structure of the ancestral complex was very similar to the human LSD1-RCOR1 complex with a root-mean-square deviation (RMSD) of 0.433 Å for 707 Cα atoms (Fig. 8c). To investigate the interaction interfaces between ancestral LSD1 and RCOR we once again subdivided the tower-linker structure into four regions (I, II, III and IV). Region III was not considered for

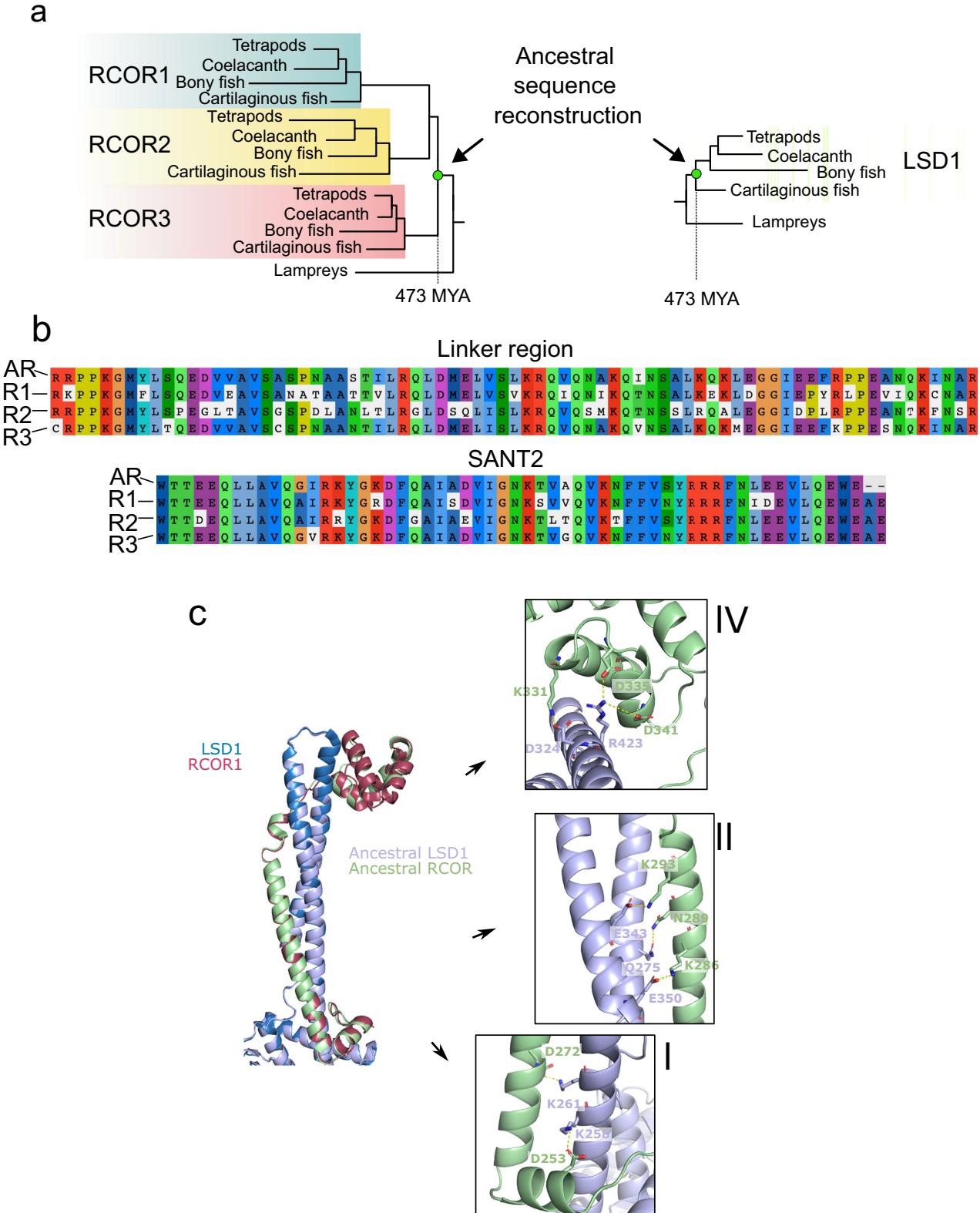

**Fig. 8 Predicted molecular interaction between the jawed vertebrate ancestral LSD1 and RCOR proteins. a** Graphical representation of the ancestral RCOR and LSD1 protein sequence reconstructions. **b** Sequence alignments of the linker region and SANT2 domain of the putative jawed vertebrate ancestral RCOR (AR), with human RCOR1 (R1), RCOR2 (R2), and RCOR3 (R3). **c** Superposition of the 3D structure of the interacting surfaces of the human RCOR1-LSD1 complex with the predicted 3D structure of the ancestral RCOR-LSD1 complex. I, II and IV highlight key amino acids for interaction in the ancestral complex.

analysis for the reason stated above. All residues analyzed below (see also Supplementary Fig. 3) are conserved or replaced by chemically similar amino acids in the ancestral protein when compared to RCOR1. Similar to the human LSD1-RCOR1 complex, region I included hydrophobic interactions as well as two salt bridges between Lys258 and Lys261 of LSD1, and Asp253 and Asp272 of RCOR (Fig. 8c, right, bottom). Also, like RCOR1, region II contains hydrophobic contacts; hydrogen bonds and salt bridges between Gln-Asn and Glu-Lys residues, respectively (Fig. 8c, right, middle). Finally, region IV also displays conserved ionic interactions between Lys-Asp and Arg-Asp residues (Fig. 8c, right, top). Overall, this shows that the gross architecture and the details of the interaction between the ancestral LSD1-RCOR heterodimer are nearly identical to the human LSD1-RCOR1 and LSD1-RCOR3 complexes.

## Methods

**Protein sequence collection and phylogenetic analyses.** We obtained lysine-specific demethylase 1 (LSD1), lysine-specific demethylase 2 (LSD2), REST Corepressor 1 (RCOR1), REST Corepressor 2 (RCOR2), and REST Corepressor 3 (RCOR3) protein sequences from the Ensembl v.102[54] using orthology and paralogy estimates from the EnsemblCompara database[55]; these estimates were obtained from an automated pipeline that considers both synteny and phylogeny to generate orthology mappings. Further, we also retrieved sequences from the National Center for Biotechnology Information (NCBI)[56] using the human (Homo sapiens) sequence as the reference for protein-BLAST (blastp)[57] against the non-redundant database (nr) with default parameters. In each case, we corroborated the presence of all described protein domains (SWIRM/AOD for LSD, ELM2/SANT1/SANT2 for RCOR).

We implemented two types of analyses that involved different sampling strategies. The first analysis aimed at understanding the evolutionary history of these groups of genes in vertebrates, and our taxonomic sampling included representative species of all main groups of vertebrates. The second analysis aimed to investigate the evolution of these genes in metazoans, thus our sampling included species of all main groups of animals. Accession numbers and details about the taxonomic sampling are available in Supplementary Data 1–4. We used the software MAFFT v.7[58] to align amino acid sequences allowing the program to choose the alignment strategies (L-INS-i for LSD alignment in vertebrates and metazoa; FFT-NS-i for RCOR in vertebrates and metazoa). We estimate phylogenetic relationships using maximum-likelihood (ML) approach as implemented in IQ-Tree v1.6.12[59]. We used the proposed model tool of IQ-Tree v.1.6.12[60] to select the best-fitting models of amino acid substitution, which selected JTT+I+G4 for LSD in vertebrates and RCOR in vertebrates and metazoans. For LSD in metazoans, the model selected was LG+F+R10. We assessed the node support using the Shimodaira–Hasegawa approximate likelihood-ratio test[61], approximate Bayes test[61,62], and ultrafast bootstrap approximation with 1000 pseudoreplicates[63,64]. We repeated each phylogenetic estimation ten times to explore the tree space, and the tree with the highest likelihood score was chosen. Sequences of monoamine-oxidases A and B (MAO-A and MAO-B), and Mitotic deacetylase-associated SANT domain protein (MIDEAS) were used as outgroup for LSD1/2 and RCOR1/2/3, respectively. Trees-supporting data are available in Supplementary Data 7.

**LSD1 alternative exon annotation.** To annotate the LSD1 alternative exon, called E8a[34], in representative species of the main groups of vertebrates (Supplementary Data 5), we retrieved LSD1 sequences from NCBI[56] and Ensembl v.102 databases[54].

We manually annotated the E8a exon and regulatory sequences by comparing known sequences using the program Blast2seq v2.5[65] with default parameters using as reference the alternative exon previously described in human (Homo sapiens, CCDS53278.1), mouse (Mus musculus, XM_006539329.5), and zebrafish (Danio rerio, ENSDART00000085758.6)[38–40].

**Dot-plots.** We retrieved the chromosomal region containing the RCOR2 gene of the painted turtle (NW_007359864.1) and the corresponding syntenic region in the chicken (Chromosome 33-NC_008465.4), and New Caledonian crow (Chromosome 34-NC_045509.1) based on the location of the flanking genes (NAA40 and MARK2). We aligned RCOR2 syntenic regions using PipMaker[66].

**Domain conservation analysis.** We constructed four amino acid multiple sequence alignments (MSA) corresponding to LSD1, RCOR1, RCOR2, and RCOR3, including representative species of all main groups of jawed vertebrates. Alignments were obtained using MAFFT v.7[58], allowing the program to choose the alignment strategies (L-INS-i, in all cases). Then, using the MSAs and PDB codes for the 3D structure of each protein (if available) as input, we estimated normalized conservation scores for each alignment independently using the ConSurf WebServer. ConSurf takes a multiple sequence alignment and builds a phylogenetic tree using a neighbor-joining algorithm. Positional conservation scores are then calculated employing empirical Bayesian methods[67,68]. The outputs were treated as continuous conservation scores which were used to construct Fig. 5 or divided into discrete categories and projected onto the 3D structure of the proteins for visualization in Fig. 6[41–43]. Protein domain positions were inferred using the human proteins with the InterPro web server[44].

**Ancestral sequence reconstructions.** We retrieved LSD1, LSD2, RCOR1, RCOR2, and RCOR3 coding sequences from NCBI[56] using human (Homo sapiens), chicken (Gallus gallus), or zebrafish (Danio rerio) sequences as references for blastp searches[57] against the non-redundant database (nr) with default parameters. Our taxonomic sampling included representative species of all main groups of chordates (mammals, birds, reptiles, bony fish, cartilaginous fish, cyclostomes, tunicates, and cephalochordates. Supplementary Data 6). Amino acid sequences were aligned using MAFFT v.7[58] allowing the program to choose the alignment strategy (L-INS-i). Nucleotide alignments were generated using the amino acid alignments as templates using PAL2NAL[69]. Ancestral sequence reconstruction was performed using IQ-Tree v1.6.12[59], including an organismal phylogenetic tree based on the most updated hypothesis for the species included in our taxonomic sampling[70–73].

**Ancestral complex structure prediction.** To predict the structure of the ancestral RCOR-LSD1 complex we used the multimer prediction tool of ColabFold[52]. We used the reconstructed ancestral sequences as a query using the structure of the RCOR1-LSD1 complex (PDB: 2V1D) as a template. The multiple sequence alignment was constructed using MMseqs2 (UniRef100 + Environmental) as provided. Input parameters were as follows: relax using amber (no), pair mode (paired + unpaired), model type (alphafold2 multimer v2), number of recycles (6), pairing strategy (greedy), max. msa (auto), num. seeds (1), use dropout (no). We selected the model with the best-predicted local-distance difference test scores (pLDDT) in the interaction interface of the complex and the best-predicted alignment error (PAE) (Supplementary Fig. 7).

**Statistics and reproducibility**. To compare the conservation scores of each protein domain (Fig. 6 and Supplementary Fig. 4) with every other domain found in the same protein, we performed the Kruskal–Wallis nonparametric test with the posterior Dunn's multiple comparisons test using Prism 9.0. We report the $p$ value of each comparison in the corresponding section of the manuscript. To compare expected and observed frequencies of highly conserved residues in LSD1's tower domain and RCOR1,3 linker domains (Fig. 7) we performed $X^2$ statistics without Yates correction. We report the resulting statistic parameters as $X^2$ (degrees of freedom, $N$ = sample size) = $X^2$ statistic value, $p = p$ value. As expected frequencies we calculated ($1/9 \times N$), where $N$ represents the number of residues in a given segment of the protein. Observed frequencies were calculated using discrete conservation scores rendered by the ConSurf output file.

**Reporting summary**. Further information on research design is available in the Nature Portfolio Reporting Summary linked to this article.

## Data availability

Command lines used for phylogenetic analysis, LSD1 alternative exon annotation, and ancestral reconstruction are available in Supplementary Data 1–6. Trees-supporting data are available in Supplementary Data 7.

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

## Acknowledgements

This work was supported by Fondo Nacional de Desarrollo Científico y Tecnológico from Chile (FONDECYT 1210471) and Millennium Nucleus of Ion Channel Associated Diseases (MiNICAD), Iniciativa Científica Milenio, Ministry of Economy, Development and Tourism from Chile to J.C.O., Fondo Nacional de Desarrollo Científico y Tecnológico from Chile (FONDECYT 1191152) to M.E.A. and Postdoctorado ANID 3230704 to M.O.-C. The authors acknowledge David Baker and Ian Humphreys, University of Washington, for their help on analyses using Gremlin. J.C.O. wants to acknowledge the members of the Integrative Biology Group, for their constant support, scientific enthusiasm, and creative feedback. We also want to thank Fabian Rentzsch for improving an early version of our work.

## Author contributions

All authors contributed extensively to the work presented in this paper. J.C.O. and M.E.A. designed and supervised the project, analyzed data, and wrote the paper. M.O.-C., G.M.O., and D.V.-V. designed and performed experiments, analyzed data, and wrote the paper. D.A. analyzed data and wrote the paper. M.P.G. performed cloning and Co-IP experiments.

## Competing interests

The authors declare no competing interests.
