## [Peer review file · Communications Biology]

Reviewers' comments:

Reviewer #1 (Remarks to the Author):

The authors analyze two families of proteins, Lysine-demethylase 1 (LSD1) and REST Corepressor (RCOR), which are known to interact in jawed vertebrates and in fruit fly as part of chromatin regulation mechanism. They look at the phylogenetic distribution of genes encoding these proteins and conclude that the three jawed vertebrate RCORs result from paralogs derived from early jawed vertebrate genome duplications. Jawless vertebrate and invertebrate animals, as well as fungi and plants have a single RCOR gene. The LSD1 genes originate from more ancient duplications. The authors outline the evolution of LSD1 exon structure and LSD1/RCOR domain structure in vertebrates. They look at amino acid conservation to predict interactions of LSD1 and RCOR proteins. They carry out immunoprecipitation to look at interactions of reconstructed ancestral jawed vertebrate LSD1 and RCOR proteins and show that these interact. Finally, they use a similar IP strategy to show that the hop plant LSD1 and RCOR proteins also interact. Thus they show that interaction among these factors is ancestral to many eukaryotes.

The paper summarizes a considerable amount of phylogenetic data about the presence/absence, multiplicity, and protein domain structure encoded by LSD1 and RCOR genes across many eukaryotic groups. It should be of interest to scientists interested in these chromatin modification systems in particular and more generally those interested in protein gain and loss. Below are a few suggestions that may clarify some points.

1. In the abstract, it would help to define the common hop as a plant ("hop plant") since the name is confusing out of context.

2. A current understanding of vertebrate genome history suggests that one of the two rounds of duplication took place prior to the divergence of agnathans and gnathostomes while the second took place in a gnathostome common ancestor (Nakatani et al. 2021, Simakov et al. 2020 referenced in manuscript). Do the RCOR findings in the manuscript imply that the presence of only one RCOR gene in the extant agnathans results from loss of a duplicate in a common cyclostome ancestor?

3. Can the authors say anything more about the loss of the RCOR/LSD1 genes in certain groups of organisms? Does this correspond to changes in other aspects of gene regulatory control?

4. Page 8: The text seems to imply that RCOR is present in most animals but it seems to be missing in a great number of phyla (or subgroups). The text should more explicitly describe the proportion of phyla that have RCOR or LSD1 genes.

5. Page 8 bottom: Although its true that transcription of a gene does not guarantee protein production, the presence of an intact gene in an animal certainly implies that protein is also present at some point in the organism's life history.

6. For the IP experiments, it would be reassuring to see that the reciprocal pull down also captures the complex and possibly that the interaction can be disrupted also by specific modifications to the presumptive interaction regions in the constructs.

Reviewer #2 (Remarks to the Author):

Summary: In this paper, the authors interrogate the evolutionary history of LSD1 and its interacting RCOR proteins in metazoans, fungi, and plants. They find that the RCOR proteins are ohnologs that arose in the last common ancestor of jawed vertebrates after their divergence from jawless vertebrates, and that LSD1/LSD2 are present in all animals and likely duplicated prior to the divergence of animals from plants. They determined that RCOR and LSD genes largely coexist within most lineages, consistent with their known interaction. Using sequence analysis, they found evidence suggesting that the micro exon of LSD1 that is specific to the neural form is confined to

mammals, turtles, and fish, consistent with past reports, and that the neuroLSD1 isoform is likely not neural restricted in fish due to an absence of the splicing regulatory sequences in these species. In addition to investigating the genes themselves, the authors also sought to determine whether residues necessary for LSD1/RCOR interaction were conserved across the paralogs and across species for each paralog, and found that residues in the linker domain of RCOR and the tower domain of LSD1, which are responsible for RCOR/LSD1 interaction, generally show higher conservation than other residues within the linker/tower domains, which otherwise appear to be less conserved than other domains within each protein. Finally, using ancestral reconstruction and CoIP experiments, the authors determine that the RCOR/LSD1 interaction was likely preserved in the ancestor of jawed vertebrates. Further, by analyzing RCOR/LSD1 homologs in the common hop, they demonstrate that this interaction is likely even older, predating the split between animals, fungi, and plants.

General Comments: Overall, the paper is well written, and presents new insight into the evolution of LSD1 and RCOR proteins and their interactions. Although the majority of the first half of the paper is largely confirmatory (which the authors acknowledge), it extends past studies by investigating LSD1/RCOR evolution in a broader range of species and provides preliminary evidence that the interaction between these two proteins is older than previously appreciated. Because of this, I believe this paper provides enough additional insight to warrant publication in *Communications Biology*. However, prior to publication, there are a few minor issues to be addressed and some experiments I would like to see that would strengthen the author's arguments about the importance of key residues/domains to the LSD1/RCOR interaction. Based on their conservation analysis of LSD1 and RCOR1/3 proteins and the interaction interfaces between them across species, the authors make several points. They find that many amino acid residues that mediate the LSD1/RCOR interaction are highly conserved across species, and thus are likely to be important for LSD1/RCOR interaction. However, given their coIP assay, it would be very easy to test these directly by repeating the CoIP experiments with varying LSD1 and RCOR proteins that have one or more of these key residues mutated. Further, it would be interesting to test if any of the conserved outwardly facing residues truly are unnecessary for LSD1/RCOR interaction by performing the CoIP interaction with mutants of these residues. The authors also propose that the coiled-coil region in the AO domain of the common hop is likely to be a Tower domain and is the mediator of its interaction with RCOR – again this is something that could be easily tested directly in CoIP assays where either the putative Tower domain is deleted or an LSD1-like ortholog in other species that naturally lacks the Tower domain (such as in *A. thaliana*). For both additional coIP assays, positive results would be quite interesting and mechanistically insightful, and negative results would serve as valuable negative controls for the CoIP assays, which the paper currently lacks.

While the evolutionary analyses are overall robust there are a few clarifications and additions that I think would make the conclusions easier to understand. First, I think it would be worthwhile to include the cartoon diagram of the RCOR/LSD1 domain structures earlier in the manuscript or at least reference Fig. 8 earlier, which will make it easier for the reader to remember the important domains. Second, while the authors provide extensive tables in the supplement on the presence of various RCOR homologs/orthologs in vertebrates and invertebrate species, the tables within the supplement lack information that is stated in the text on absence of RCOR/LSD1. The authors should include this information within the tables to improve clarity. Additionally, I think the paper would benefit from having a similar figure (in the main body or supplement) as Fig. 4A for invertebrates, including a phylogenetic tree with phyletic information layered on.

Regarding the search for RCOR/LSD1 homologs in more distantly related species such as fungi and plants, the authors use the human protein sequence as a query. While this is adequate to pull out a few additional candidates, human is so far diverged from fungi and plants that it is unlikely to pull out most homologs by sequence similarity alone. Thus, it would be worthwhile to take the fungal and plant sequences that are initially identified by this search and to use them as a query to see if any additional homologs can be identified in closely related genomes.

Finally, the authors speculate that the lower conservation of residues within the linker/Tower domains relative to other functional domains in their respective proteins might be reflective of diversification of RCOR orthologues and may mediate their interaction with distinct proteins. However, it seems plausible that the lower conservation is simply due to a reduced selection pressure on those residues relative to residues that are responsible for the linker/Tower interaction. The authors should either acknowledge this possibility or provide additional analyses

supporting their hypothesis.

Specific Comments:

- Regarding the phyletic analyses comparing coexistence of LSD1 with RCOR in various lineages, it would be worthwhile to clarify why the authors are confident that an apparent “absence” of one of the two interactors is a true absence vs. absent due to technical errors (poor genome, assembly gaps, etc)
- Page 10, SRS analysis – here the authors conclude that because the SRS is not present in fish species that LSD1 is not likely to be restricted to neural lineages in fish. While that seems to be supported from RNA-seq in zebrafish, is it possible that the SRS sequence is there but just sufficiently diverged as to be undetectable?
- Page 12, LSD1/RCOR residue conservation analysis – here the authors state that 54.7% and 76.5% of the deeply conserved residues for LSD1 and RCOR1 respectively are directly involved in the interaction interface, and imply that this is a large number. However, they do not provide any sort of null expectation, which would make the results more interpretable. It might be more interesting to ask if there are any residues at the interface between LSD1/RCOR1 that are not deeply conserved

Reviewer #3 (Remarks to the Author):

LSD1 is a histone demethylase known to interact with REST Corepressor proteins – RCOR – to induce the repression of target loci during development. Vertebrate genomes are known to encode 2 LSD, and 3 RCOR genes and the present study seeks to understand when and how LSD1/RCOR interaction evolved.

First, the authors used phylogenetic approaches to identify the evolutionary trajectories LSD1 and RCOR genes. They confirmed an early duplication of LSD1/LSD2 in eukaryotes and found that RCOR duplicated into 3 copies following rounds of whole genome duplication in the last common ancestor of all jawed vertebrates. The authors suggest that this amplification allowed for RCOR sub-functionalization. They went on to identify lineage-specific event of LSD1 or RCOR loss; raising the question of their essential functions in these species (e.g. birds RCOR2 loss; and neuron specific LSD1-8a).

Second, they explored the evolution of the molecular interaction between LSD1/RCOR. They found that overall, interacting domains are not the most conserved portion of both proteins. However, structural modeling showed that most interface residues are indeed conserved between paralogs and orthologs.

Finally, using heterologous expression of tagged proteins in human cells, the authors found interactions between reconstructed ancestral LSD1/RCOR of jawed vertebrates and between LSD1 and RCOR of the common hop. The authors conclude that the LSD1/RCOR interaction arose deep in the eukaryotic lineage, probably 1.4 billion years ago, in the last common ancestor of plants and animals.

Overall, I found this study technically sound and the methods used are appropriate. The manuscript is well written and the conclusion are supported by the data presented. This present novel findings of interests to chromatin, genome and evolutionary biologist alike.

I have a few comments detailed below, in no particular order:

- 1) It seems that there are some studies investigating RCOR3 – notably as a competitive inhibitor of RCOR1 and 2, e.g. the article cited by the authors PMID: 24843136. Maybe the authors can discuss this specific function of RCOR3 in the context of their study.
- 2) When discussing the lineage-specific regulation of neuronal LSD1-8a, the authors could investigate EST or RNA-seq datasets to support their hypothesis.
- 3) Can the author provide more details, in the method section and main text, on the conservation score provided by ConSurf.
- 4) Related to Figure 5, Result section: Can the author provide some statistical support for the comparison of domain conservation scores (maybe a chi-square (?)).
- 5) A similar comment can be made when discussing the proportion of domains with conserved

residues – can the authors say if the % they observe is higher or lower than what would be expected by chance.

6) Can the author detail the rules underlying ancestor sequence reconstruction? Is it solely based on parsimony? Which species were used and why?

Rebuttal Letter

Dear Reviewers:

We appreciate your feedback. Your comments greatly contribute to improving our manuscript. Below, we respond to each of your comments.

Reviewer #1 (Remarks to the Author):

The authors analyze two families of proteins, Lysine-demethylase 1 (LSD1) and REST Corepressor (RCOR), which are known to interact in jawed vertebrates and in fruit fly as part of chromatin regulation mechanism. They look at the phylogenetic distribution of genes encoding these proteins and conclude that the three jawed vertebrate RCORs result from paralogs derived from early jawed vertebrate genome duplications. Jawless vertebrate and invertebrate animals, as well as fungi and plants have a single RCOR gene. The LSD1 genes originate from more ancient duplications. The authors outline the evolution of LSD1 exon structure and LSD1/RCOR domain structure in vertebrates. They look at amino acid conservation to predict interactions of LSD1 and RCOR proteins. They carry out immunoprecipitation to look at interactions of reconstructed ancestral jawed vertebrate LSD1 and RCOR proteins and show that these interact. Finally, they use a similar IP strategy to show that the hop plant LSD1 and RCOR proteins also interact. Thus they show that interaction among these factors is ancestral to many eukaryotes.

The paper summarizes a considerable amount of phylogenetic data about the presence/absence, multiplicity, and protein domain structure encoded by LSD1 and RCOR genes across many eukaryotic groups. It should be of interest to scientists interested in these chromatin modification systems in particular and more generally those interested in protein gain and loss. Below are a few suggestions that may clarify some points.

1. In the abstract, it would help to define the common hop as a plant (“hop plant”) since the name is confusing out of context.

We appreciate the reviewer's comment but decided to eliminate the analyses performed on plants and fungi from the manuscript. This is because our phylogenetic analyses recovered RCOR from plants and fungi in the same clade as arthropods. Under this scenario, we investigated if this could be due to plant and fungi material contamination with arthropod sequences. We reasoned that if we couldn't find RCOR sequences in the plant-fungi genomes, this would reflect contamination. Accordingly, we found RCOR plant and fungi sequences only in transcriptomic databases using tBLASTn, specifically TSA (Transcriptome Shotgun Assembly). Still, no plant sequences could be retrieved from genomic sequences (Refseq Genome Database, Whole Genome Shotgun Contigs, High Throughput Genomic Sequences). We analyzed the five plant and one fungi species included in our manuscript. The most likely explanation for our findings is that the RCOR plant and fungi transcript sequences we retrieved are products of transcriptomic contamination from arthropods. The situation we described above was also independently noticed by Dr. Fabian Rentzsch, who kindly emailed us about this situation. So, we confidently feel that the sequences

we retrieved from plants and fungi are the product of contamination and must not be included in the manuscript.

2. A current understanding of vertebrate genome history suggests that one of the two rounds of duplication took place prior to the divergence of agnathans and gnathostomes while the second took place in a gnathostome common ancestor (Nakatani et al. 2021, Simakov et al. 2020 referenced in manuscript). Do the RCOR findings in the manuscript imply that the presence of only one RCOR gene in the extant agnathans results from loss of a duplicate in a common cyclostome ancestor?

The reviewer is correct. Assuming that one of the two rounds of whole-genome duplication occurred before the divergence of agnathans and gnathostomes indicates that the ancestor of vertebrates had two RCOR genes. Agnathans and gnathostomes inherited this repertoire. One of the copies was lost in the first group, while in the ancestor of gnathostomes, the second round of whole-genome duplication produced four RCOR copies, and subsequently, one copy was lost. To make this clear, we added more information. The new passage reads as follows: "It is widely accepted that the evolution of vertebrates was shaped by ancient whole-genome duplications (WGDs) (Meyer and Schartl 1999; McLysaght et al. 2002; Dehal and Boore 2005; Hoegg and Meyer 2005; Putnam et al. 2008). Although the most accepted hypothesis invokes two rounds of WGD during the evolutionary history of vertebrates, the timing of these duplication events is still a matter of debate (Simakov et al. 2020; Nakatani et al. 2021). The most recent hypothesis suggests that one of the duplications occurred before the divergence of cyclostomes and gnathostomes, while the second took place in the gnathostome ancestor (Simakov et al. 2020; Nakatani et al. 2021). This scenario suggests that the vertebrate ancestor had two RCOR genes, a gene repertoire inherited by cyclostomes and gnathostomes. One of the copies was lost in the first group, while in the ancestor of gnathostomes, the second round of whole-genome duplication produced four RCOR copies, and subsequently, one copy was lost."

3. Can the authors say anything more about the loss of the RCOR/LSD1 genes in certain groups of organisms? Does this correspond to changes in other aspects of gene regulatory control?

The scarcity of studies focusing on gene expression regulation in organisms with either RCOR or LSD1 makes it challenging to derive conclusions regarding the regulatory consequences of gene loss. This is why we did not speculate about it in the manuscript. However, studies on plants (*Arabidopsis*), which has four LSD1 homologs and no RCOR, show the involvement of these proteins in the regulation of flowering by controlling H3K4me2 levels in target genes (Jiang et al., 2007). Although this evidence is indirect, it is remarkable that the substrate specificity of LSD1 homologs is retained in plants. Additionally, these results suggest that H3K4me2 *in vivo* demethylation capabilities in *Arabidopsis* are not RCOR dependent, as opposed to the case in mammals. Thus, either *Arabidopsis* LSD1 homologs alone interact with the chromatin appropriately so that demethylation can occur, or another protein is functioning as a bridge between LSD1 homologs and chromatin in this organism.

Jiang D, et al. Plant Cell. 2007;19:2975–87 (PMID: 17921315).

4. Page 8: The text seems to imply that RCOR is present in most animals but it seems to be missing in a great number of phyla (or subgroups). The text should more explicitly describe the proportion of phyla that have RCOR or LSD1 genes.

We appreciate this comment. We included quantitative information for each gene in the current manuscript version to solve this problem. The new passage reads as follows: “In invertebrates, we found RCOR genes in 19 of 35 phyla analyzed. Specifically, RCOR was found in all analyzed phyla but in Tardigrada, Kinorhyncha, Loricifera, Nematomorpha, Sipuncula, Orthonectida, Micrognathozoa, Rhombozoa, Onychophora, Gnathostomulida, Gastrotricha, Entoprocta, Cycliophora, Chaetognatha, Acanthocephala, and Entoprocta (Table S2). LSD1 and LSD2 genes were found in all main vertebrate groups (Fig. 3 and Table S3). LSD1 was found in 22 of 34 phyla studied, excluding the invertebrates phyla Kinorhyncha, Loricifera, Nematomorpha, Sipuncula, Orthonectida, Micrognathozoa, Onychophora, Gnathostomulida, Gastrotricha, Entoprocta, Cycliophora, and Chaetognatha. On the other hand, LSD2 was not found in some Arthropods, as previously reported (Zhou and Ma 2008), and Placozoa, Tardigrada, Kinorhyncha, Loricifera, Nematomorpha, Sipuncula, Rotifera, Platyhelminthes, Orthonectida, Micrognathozoa, Rhombozoa, Onychophora, Gnathostomulida, Gastrotricha, Entoprocta, Cycliophora, Chaetognatha, and Acanthocephala (Table S4).”

5. Page 8 bottom: Although its true that transcription of a gene does not guarantee protein production, the presence of an intact gene in an animal certainly implies that protein is also present at some point in the organism’s life history.

The reviewer is correct. We meant to be cautious in suggesting that LSD1 is functional in the water bear by only verifying the presence of the gene. To solve this problem, we removed the following statement: “Bearing in mind that identifying a gene does not necessarily mean that the protein is expressed”. Thus, the new paragraph reads: “The results of our analysis indicate that there is a greater abundance of species in which RCOR genes co-exist with LSD1, as compared to species in which only one of these genes is present (Table S2 and S4). Nevertheless, there exist certain instances that deviate from the norm. For example, the organism known as the water bear (*Ramazzottius varieornatus*) exhibits the presence of LSD1 but not RCOR. This example would imply that LSD1 is functionally independent of RCOR in some animal lineages. This species and others could provide a suitable platform to investigate RCOR (or LSD1) functions that are LSD1 (or RCOR)-independent to further understand the diverse range of biological processes in which these proteins participate.”

6. For the IP experiments, it would be reassuring to see that the reciprocal pull down also captures the complex and possibly that the interaction can be disrupted also by specific modifications to the presumptive interaction regions in the constructs.

We regret to inform you that despite numerous attempts to address the reviewers' requirements regarding the interaction between ancestral LSD1 (corresponding to amino acids Val-174 to Met-852 of human LSD1) and RCOR (corresponding to amino acids Gly-103 to Glu-441 of human RCOR1), we could not obtain complete co-immunoprecipitation data.

Initially, control assays revealed a non-specific interaction between the Myc epitope and LSD1, leading us to discard the results presented in the former Figure 7C. Consequently, we synthesized new plasmids encoding ancestral and human RCOR1, tagged with the FLAG epitope, in collaboration with Twist Biosciences and Genescript companies. Surprisingly, the recombinant proteins displayed considerable instability, further exacerbated when they were co-transfected. As shown in **Figure 1**, the Western blot lanes (1, 2, 3, and 4) did not show the expected signal size for ancestral and human HA-LSD1 (signal at 75 kDa) or Flag-RCOR (expected size 45 kDa). However, ancestral and human HA-LSD1 were detected (although weak) when co-transfected with the control Flag-GFP (signal at 35 kDa).

Figure 1: Western blot of whole cell extracts of HEK293 co-transfected with ancestral (A) or human (H) HA-LSD1 and either Flag-RCOR or Flag-GFP were SDS-PAGE fractionated and recombinant proteins recognized by anti-HA (first) and anti-flag (second).

Despite our extensive efforts, the limited abundance of the recombinant proteins made it impossible to obtain reliable data on the interaction between ancestral LSD1 and RCOR1 through Co-IP experiments. Nevertheless, we confirmed the interaction between endogenous human LSD1 and ancestral RCOR, as depicted in **Figure 2**. However, this data alone does not substantiate the interaction between ancestral RCOR and LSD1. Consequently, we have made the decision to exclude the co-immunoprecipitation data from the manuscript.

Figure 2: Ancestral RCOR interacts with endogenous human LSD1. Whole-cell extracts of HEK293 transfected with ancestral Flag-RCOR (A, B) or Flag-GFP (C) were immunoprecipitated with anti-LSD1 (rabbit polyclonal, Abcam). Immunoprecipitates were SDS-PAGE fractionated, and Western blots were revealed with anti-Flag (A, C) and anti-LSD1 (B, after anti-Flag).

Lastly, the low levels of the recombinant protein LSD1 suggest that this protein's N- and C-terminals play crucial roles in their stability. However, investigating this matter is beyond the scope of our current work, as synthesizing plasmids encoding full-length proteins would be financially prohibitive.

We hope you understand our choice of removing the results related to the physical interaction of ancestral proteins while emphasizing the valuable insights gained through software-based modeling.

Reviewer #2 (Remarks to the Author):

Summary: In this paper, the authors interrogate the evolutionary history of LSD1 and its interacting RCOR proteins in metazoans, fungi, and plants. They find that the RCOR proteins are ohnologs that arose in the last common ancestor of jawed vertebrates after their divergence from jawless vertebrates, and that LSD1/LSD2 are present in all animals and likely duplicated prior to the divergence of animals from plants. They determined that RCOR and LSD genes largely coexist within most lineages, consistent with their known interaction. Using sequence analysis, they found evidence suggesting that the micro exon of LSD1 that is specific to the neural form is confined to mammals, turtles, and fish, consistent with past reports, and that the neuroLSD1 isoform is likely not neural restricted in fish due to an absence of the splicing regulatory sequences in these species. In addition to investigating the genes themselves, the authors also sought to determine whether residues necessary for

LSD1/RCOR interaction were conserved across the paralogs and across species for each paralog, and found that residues in the linker domain of RCOR and the tower domain of LSD1, which are responsible for RCOR/LSD1 interaction, generally show higher conservation than other

residues within the linker/tower domains, which otherwise appear to be less conserved than other domains within each protein. Finally, using ancestral reconstruction and CoIP experiments, the authors determine that the RCOR/LSD1 interaction was likely preserved in the ancestor of jawed vertebrates. Further, by analyzing RCOR/LSD1 homologs in the common hop, they demonstrate that this interaction is likely even older, predating the split between animals, fungi, and plants.

General Comments: Overall, the paper is well written, and presents new insight into the evolution of LSD1 and RCOR proteins and their interactions. Although the majority of the first half of the paper is largely confirmatory (which the authors acknowledge), it extends past studies by investigating LSD1/RCOR evolution in a broader range of species and provides preliminary evidence that the interaction between these two proteins is older than previously appreciated. Because of this, I believe this paper provides enough additional insight to warrant publication in *Communications Biology*. However, prior to publication, there are a few minor issues to be addressed and some experiments I would like to see that would strengthen the author's arguments about the importance of key residues/domains to the LSD1/RCOR interaction.

- 1. Based on their conservation analysis of LSD1 and RCOR1/3 proteins and the interaction interfaces between them across species, the authors make several points. They find that many amino acid residues that mediate the LSD1/RCOR interaction are highly conserved across species, and thus are likely to be important for LSD1/RCOR interaction. However, given their coIP assay, it would be very easy to test these directly by repeating the CoIP experiments with varying LSD1 and RCOR proteins that have one or more of these key residues mutated. Further, it would be interesting to test if any of the conserved outwardly facing residues truly are unnecessary for LSD1/RCOR interaction by performing the CoIP interaction with mutants of these residues.**

The reviewer is correct in mentioning that mutagenesis experiments could confirm our observations that highly conserved residues in the interface between both proteins are likely to aid LSD1-RCOR interaction. However, we think this is out of the scope of our manuscript.

- 2. The authors also propose that the coiled-coil region in the AO domain of the common hop is likely to be a Tower domain and is the mediator of its interaction with RCOR – again this is something that could be easily tested directly in CoIP assays where either the putative Tower domain is deleted or an LSD1-like ortholog in other species that naturally lacks the Tower domain (such as in *A. thaliana*). For both additional coIP assays, positive results would be quite interesting and mechanistically insightful, and negative results would serve as valuable negative controls for the CoIP assays, which the paper currently lacks.**

We appreciate the reviewer's comment but decided to eliminate the analyses performed on plants and fungi from the manuscript. This is because our phylogenetic analyses recovered RCOR from plants and fungi in the same clade as arthropods. Under this scenario, we investigated if this could be due to plant and fungi material contamination with arthropod sequences. We reasoned that if we couldn't find RCOR sequences in the plant-fungi genomes, this would reflect contamination. Accordingly, we found RCOR plant and fungi sequences only in transcriptomic databases using

tBLASTn, specifically TSA (Transcriptome Shotgun Assembly). Still, no plant sequences could be retrieved from genomic sequences (Refseq Genome Database, Whole Genome Shotgun Contigs, High Throughput Genomic Sequences). We analyzed the five plant and one fungi species included in our manuscript. The most likely explanation for our findings is that the RCOR plant and fungi transcript sequences we retrieved are products of transcriptomic contamination from arthropods. The situation we described above was also independently noticed by Dr. Fabian Rentzsch, who kindly emailed us about this situation. So, we confidently feel that the sequences we retrieved from plants and fungi are the product of contamination and must not be included in the manuscript.

While the evolutionary analyses are overall robust there are a few clarifications and additions that I think would make the conclusions easier to understand. First, I think it would be worthwhile to include the cartoon diagram of the RCOR/LSD1 domain structures earlier in the manuscript or at least reference Fig. 8 earlier, which will make it easier for the reader to remember the important domains.

The reviewer is correct. To solve this problem, we included a diagram of the RCOR/LSD1 domain structures earlier in the manuscript. This cartoon diagram is now presented in Figure 1.

Second, while the authors provide extensive tables in the supplement on the presence of various RCOR homologs/ohnologs in vertebrates and invertebrate species, the tables within the supplement lack information that is stated in the text on absence of RCOR/LSD1. The authors should include this information within the tables to improve clarity.

Additionally, I think the paper would benefit from having a similar figure (in the main body or supplement) as Fig. 4A for invertebrates, including a phylogenetic tree with phyletic information layered on.

The reviewer is correct. We now included information about the absence of RCOR and LSD1 in different animal phyla. This can be found in supplementary Tables S2 and S4.

To be consistent, we made a figure similar to 4A, now showing our results for invertebrates. This is now called Figure S2.

Regarding the search for RCOR/LSD1 homologs in more distantly related species such as fungi and plants, the authors use the human protein sequence as a query. While this is adequate to pull out a few additional candidates, human is so far diverged from fungi and plants that it is unlikely to pull out most homologs by sequence similarity alone. Thus, it would be worthwhile to take the fungal and plant sequences that are initially identified by this search and to use them as a query to see if any additional homologs can be identified in closely related genomes.

We appreciate the reviewer's comment but decided to eliminate the analyses performed on plants and fungi from the manuscript. This is because our phylogenetic analyses recovered RCOR from plants and fungi in the same clade as arthropods. Under this scenario, we investigated if this could

be due to plant and fungi material contamination with arthropod sequences. We reasoned that if we couldn't find RCOR sequences in the plant-fungi genomes, this would reflect contamination. Accordingly, we found RCOR plant and fungi sequences only in transcriptomic databases while using tBLASTn, specifically TSA (Transcriptome Shotgun Assembly). Still, no plant sequences could be retrieved from genomic sequences (Refseq Genome Database, Whole Genome Shotgun Contigs, High Throughput Genomic Sequences). We analyzed the five plant and one fungi species included in our manuscript. The most likely explanation for our findings is that the RCOR plant and fungi transcript sequences we retrieved are products of transcriptomic contamination from arthropods. The situation we described above was also independently noticed by Dr. Fabian Rentzsch, who kindly emailed us about this situation. So, we confidently feel that the sequences we retrieved from plants and fungi are the product of contamination and must not be included in the manuscript.

Finally, the authors speculate that the lower conservation of residues within the linker/Tower domains relative to other functional domains in their respective proteins might be reflective of diversification of RCOR ohnologues and may mediate their interaction with distinct proteins. However, it seems plausible that the lower conservation is simply due to a reduced selection pressure on those residues relative to residues that are responsible for the linker/Tower interaction. The authors should either acknowledge this possibility or provide additional analyses supporting their hypothesis.

We appreciate the comment. We corrected the manuscript and recognized that as a possibility. The new passage reads as follows: “Lower mean conservation scores in the linker and tower domains could indicate a lower selective pressure on these regions relative to the other functional domains. Alternatively, linker and Tower domain variability could have made possible the flexible encounter of different RCORs’ with LSD1 and the modulation of the catalytic activity that these exert on LSD1.”

Specific Comments:

Regarding the phyletic analyses comparing coexistence of LSD1 with RCOR in various lineages, it would be worthwhile to clarify why the authors are confident that an apparent “absence” of one of the two interactors is a true absence vs. absent due to technical errors (poor genome, assembly gaps, etc)

We appreciate this comment. We addressed this by saying that a given gene or protein was “not found” instead of saying “missing” or “absent.” In some cases, we included “suggests the absence” instead of “confirms the absence,” as in, “A comparison of the chromosomal regions (flanked by NAA40 and MARK2 genes) in chicken and painted turtle suggests the absence of the RCOR2 gene in the chicken genome (Galliformes)”. This way, we explicitly recognize the possibility that a particular genomic region is either genuinely absent in the species or not accessible to our searches due to different reasons, as the reviewer points out. In the case of SRS (Splicing Regulatory Sequences) analyses, we attribute “absence” characteristics when we could not find SRS sequences consistently (e.g., all birds analyzed or all crocodiles analyzed) in

a particular group, which helped us identify technical difficulties like the ones the reviewer mentions.

Page 10, SRS analysis – here the authors conclude that because the SRS is not present in fish species that LSD1 is not likely to be restricted to neural lineages in fish. While that seems to be supported from RNA-seq in zebrafish, is it possible that the SRS sequence is there but just sufficiently diverged as to be undetectable?

We appreciate this comment. It is certainly possible that SRS-like sequences are still present in fish but that they have diverged in comparison to mammals, so we can not recognize them. However, given that the E8a-like exon is ubiquitously in zebrafish (Tamaoki et al. 2020), possible SRS-like fish sequences could have a different function than regulation of E8a splicing in neurons.

Page 12, LSD1/RCOR residue conservation analysis – here the authors state that 54.7% and 76.5% of the deeply conserved residues for LSD1 and RCOR1 respectively are directly involved in the interaction interface, and imply that this is a large number. However, they do not provide any sort of null expectation, which would make the results more interpretable. It might be more interesting to ask if there are any residues at the interface between LSD1/RCOR1 that are not deeply conserved

The reviewer correctly indicates we don't provide a null expectation to strengthen our conclusions. To fix this problem, we now specify what we should expect by chance (see methods for further details) and how our observations compare.

The new passage is as follows:

“In the case of LSD1, 41.0% of the residues of the tower domain were found to be highly conserved, which is significantly higher than expected by chance (χ^2 (df = 1, $N = 105$) = 90.4, $p < 0.0001$) (Fig. 6A, dark red). These residues are enriched in the interface between LSD1 and RCOR1 as 61.9% of interface residues are highly conserved. These amino acids are distributed along the four segments analyzed in supplementary figure 3. The other group, i.e., the highly conserved amino acids that do not participate in the interaction interface, are located mainly on the basal/middle section of the $T\alpha 2$ helix and the distal segment of the $T\alpha 1$ helix (Fig. 6A). The specific functional significance of these amino acids remains to be studied.

As for RCOR1, 33.7% of residues in its linker domain displayed highly conserved scores, this is significantly more than expectations by chance (χ^2 (df = 1, $N = 74$) = 32.4, $p < 0.0001$) (Fig. 6B). We observed enrichment of this group of amino acids in the contacts between RCOR1 and LSD1, with 59.1% of them classified as highly conserved. The majority of these residues are located in the $L\alpha 2$ helix. Interestingly, however, three highly conserved residues in the $L\alpha 2$ helix, Arg347, Gln350, and Gln354 (Fig. 6B, light blue arrows), are oriented towards the opposite direction of LSD1. The function of these amino acids is currently unknown, but given their highly conserved characteristics, they deserve further attention. Finally, in the case of RCOR3, 31.6% of the residues in its linker domain are highly conserved, which is higher than expected by chance (χ^2 (df = 1, $N = 76$) = 28.4, $p < 0.0001$). Additionally, 35% of the residues in the RCOR3-LSD1 interface are highly conserved (Fig. 6C). Furthermore, for RCOR3, highly conserved amino acids

are not solely enriched in the L α 2 helix but are more evenly distributed between the L α 1 and L α 2 helices. In addition, and as seen with RCOR1, three L α 2 residues, Gln315, Asn316, and Gln319, oriented themselves on the opposite side of the L α 2 helix, away from the interaction interface (Fig. 6C, light blue arrows).”

Reviewer #3 (Remarks to the Author):

LSD1 is a histone demethylase known to interact with REST Corepressor proteins – RCOR – to induce the repression of target loci during development. Vertebrate genomes are known to encode 2 LSD, and 3 RCOR genes and the present study seeks to understand when and how LSD1/RCOR interaction evolved.

First, the authors used phylogenetic approaches to identify the evolutionary trajectories LSD1 and RCOR genes. They confirmed an early duplication of LSD1/LSD2 in eukaryotes and found that RCOR duplicated into 3 copies following rounds of whole genome duplication in the last common ancestor of all jawed vertebrates. The authors suggest that this amplification allowed for RCOR sub-functionalization. They went on to identify lineage-specific event of LSD1 or RCOR loss; raising the question of their essential functions in these species (e.g. birds RCOR2 loss; and neuron specific LSD1-8a).

Second, they explored the evolution of the molecular interaction between LSD1/RCOR. They found that overall, interacting domains are not the most conserved portion of both proteins. However, structural modeling showed that most interface residues are indeed conserved between paralogs and orthologs.

Finally, using heterologous expression of tagged proteins in human cells, the authors found interactions between reconstructed ancestral LSD1/RCOR of jawed vertebrates and between LSD1 and RCOR of the common hop. The authors conclude that the LSD1/RCOR interaction arose deep in the eukaryotic lineage, probably 1.4 billion years ago, in the last common ancestor of plants and animals.

Overall, I found this study technically sound, and the methods used are appropriate. The manuscript is well written and the conclusion are supported by the data presented. This present novel findings of interests to chromatin, genome and evolutionary biologist alike.

I have a few comments detailed below, in no particular order:

1) It seems that there are some studies investigating RCOR3 – notably as a competitive inhibitor of RCOR1 and 2, e.g. the article cited by the authors PMID: 24843136. Maybe the authors can discuss this specific function of RCOR3 in the context of their study.

We thank the reviewer for providing this information. We included a comment on this study in the manuscript as relevant evidence for the functional diversification of RCOR proteins in vertebrates. The statement is the following: “Studies in RCOR1 null mice reveal a crucial role of this protein in erythropoiesis and the proliferation of regulatory T cells (Yao et al. 2014; Xiong et al. 2020b). On the other hand, RCOR2 is significantly expressed in embryonic stem cells, regulating their proliferation and pluripotency (Yang et al. 2011). In the case of RCOR3, an isoform without the

SANT2 domain has been shown to play antagonistic roles compared to RCOR1 and RCOR2 during myeloid cell lineage differentiation (Upadhyay et al. 2014).”

2) When discussing the lineage-specific regulation of neuronal LSD1-8a, the authors could investigate EST or RNA-seq datasets to support their hypothesis.

We appreciate this suggestion. RNA-seq experiments to detect microexons require high sequencing depth, which is not commonly available for “non-model” species. Also, we must consider that for the analysis of microexon 8a inclusion (or exclusion), we need tissue-specific databases; this eliminates the possibility of using EST or RNA-seq datasets.

Additionally, a search for 8a-aligned ESTs using the Genome Browser non-primate EST database consistently yields no matches.

(https://genome.ucsc.edu/cgi-bin/hgTracks?db=hg38&lastVirtModeType=default&lastVirtModeExtraState=&virtModeType=default&virtMode=0&nonVirtPosition=&position=chr1%3A23065588%2D23067086&hgid=1699181658_B8aKOdwXgzApQ6gxHhzAUk87AzjU).

3) Can the author provide more details, in the method section and main text, on the conservation score provided by ConSurf.

We included additional information in the main text and methods section specifying how the different scores provided by ConSurf were used to construct the figures.

The information provided in the methods is: “Then, using the MSA and PDB codes for the 3D structure of each protein (if available) as input, we estimated normalized conservation scores for each alignment independently using the ConSurf WebServer. ConSurf takes a multiple sequence alignment as an input and builds a phylogenetic tree using a neighbor-joining algorithm. Positional conservation scores are then calculated using Rate4Site employing empirical Bayesian or maximum likelihood methods (Pupko et al. 2002; Mayrose et al. 2004). The outputs were treated as continuous conservation scores, which were used to construct Fig. 5 or divided into discrete categories and projected onto the 3D structure of the proteins for visualization in Fig. 6 (Landau et al. 2005; Ashkenazy et al. 2010; Ashkenazy et al. 2016).”

In the main text, we added this paragraph according to the reviewer's request:

“To delve into the evolution of the RCOR and LSD1 molecular interaction, we studied the conservation of their functional domains in jawed vertebrates. To this end, multiple sequence alignments for each protein, incorporating representative species of all major groups of jawed vertebrates (mammals, reptiles, birds, amphibians, bony fish, and cartilaginous fish), were constructed. These alignments were used as an input for the ConSurf web server (Landau et al. 2005; Ashkenazy et al. 2010; Ashkenazy et al. 2016) to calculate positional conservation scores (Fig. 5). ConSurf outputs either continuous conservation scores, which were used to construct

Fig. 5 or creates discrete groups, ranked 1 (variable) - 9 (conserved) which were used to construct Fig. 6.”

4) Related to Figure 5, Result section: Can the author provide some statistical support for the comparison of domain conservation scores (maybe a chi-square (?)).

We included statistical analyses (Kruskal-Wallis, with Dunn's multiple comparisons test) to compare the median of each domain conservation score with every other domain in each protein. We included our findings in the manuscript in two separate paragraphs.

“When comparing each domain's conservation scores in the case of RCOR1, we found that its linker domain is significantly less conserved (higher normalized core) than SANT1 ($p < 0.0001$) and SANT2 ($p < 0.0001$). Analyses for RCOR2 rendered similar results with its linker domain significantly less conserved than SANT1 ($p < 0.0001$), SANT2 ($p = 0.0016$) and ELM2 ($p < 0.0001$). Finally, RCOR3s` linker domain was also found to be less conserved than SANT1 ($p < 0.0001$) and SANT2 ($p = 0.0003$) (Fig. 6 and Fig. S3)”.

and

“Analysis of LSD1s` functional domains revealed a similar situation to that of RCORs. The RCOR-interacting tower domain showed significantly lower conservation than the other functional domains of the protein, namely the SWIRM, amine oxidase N-terminal ($p = 0.0007$) and amine oxidase C-terminal domains ($p < 0.0001$) between which it is inserted (Fig. 6 and S3).”

5) A similar comment can be made when discussing the proportion of domains with conserved residues – can the authors say if the % they observe is higher or lower than what would be expected by chance.

The reviewer correctly points out that we don't provide a null expectation to strengthen our statements. We now specified what one would expect by chance (see methods for further details) and how our observations compare to that.

The new passage is as follows:

“In the case of LSD1, 41.0% of the residues of the tower domain were found to be highly conserved, which is significantly higher than expected by chance (X^2 ($df = 1$, $N = 105$) = 90.4, $p < 0.0001$) (Fig. 6A, dark red). These residues are enriched in the interface between LSD1 and RCOR1 as 61.9% of interface residues are highly conserved. These amino acids are distributed along the four segments analyzed in supplementary figure 3. The other group, i.e., the highly conserved amino acids that do not participate in the interaction interface, are located mostly on the basal/middle section of the $T\alpha 2$ helix and the distal segment of the $T\alpha 1$ helix (Fig. 6A). The specific functional significance of these amino acids remains to be studied.

As for RCOR1, 33.7% of residues in its linker domain displayed highly conserved scores, this is significantly more than expectations by chance (X^2 ($df = 1$, $N = 74$) = 32.4, $p < 0.0001$) (Fig. 6B). We observed enrichment of this group of amino acids in the contacts between RCOR1 and

LSD1, with 59.1% of them classified as highly conserved. The majority of these residues are located in the L α 2 helix. Interestingly, however, three highly conserved residues in the L α 2 helix, Arg347, Gln350, and Gln354 (Fig. 6B, light blue arrows) are oriented towards the opposite direction of LSD1. The function of these amino acids is currently unknown, but given their highly conserved characteristics, they deserve further attention. Finally, in the case of RCOR3, 31.6% of the residues in its linker domain are highly conserved, which is higher than expected by chance (χ^2 (df = 1, N = 76) = 28.4, $p < 0.0001$). Additionally, 35% of the residues in the RCOR3-LSD1 interface are highly conserved (Fig. 6C). Furthermore, for RCOR3, highly conserved amino acids are not solely enriched in the L α 2 helix, but are more evenly distributed between the L α 1 and L α 2 helices. In addition, and as seen with RCOR1, three L α 2 residues, Gln315, Asn316, and Gln319, oriented themselves on the opposite side of the L α 2 helix, away from the interaction interface (Fig. 6C, light blue arrows).”

6) Can the author detail the rules underlying ancestor sequence reconstruction? Is it solely based on parsimony? Which species were used and why?

Ancestral sequence reconstruction was performed using the software IQ-Tree v1.6.12. This program uses a maximum likelihood approach to infer ancestral states. To do so, we provided IQ-Tree with 1) an organismal phylogenetic tree based on the most updated hypothesis for the species included in our taxonomic sampling, 2) a nucleotide alignment of manually curated sequences, and 3) an evolutionary model of evolution, previously inferred using the model finder routine of IQ-Tree v1.6.12. After that, the ancestral sequence reconstruction is performed. Since our goal was to reconstruct the RCOR and LSD1 proteins present in the ancestor of jawed vertebrates, our taxonomic sampling included representative species of all main groups of chordates (mammals, birds, reptiles, bony fish, cartilaginous fish, cyclostomes, tunicates, and cephalochordates). Codes of protein and transcripts used for the ancestral reconstruction are detailed in supplementary table S6 (Table S6).

REVIEWERS' COMMENTS:

Reviewer #1 (Remarks to the Author):

The authors have addressed my comments with adequate responses and changes to the manuscript. The paper is should be of interest to evolutionary biologists in general and those who are more directly interested in regulatory biology.

Reviewer #2 (Remarks to the Author):

The authors have addressed all my concerns. I especially appreciate their courage to remove the plant data from the paper in light of subsequent evidence suggesting the conclusions were due to artifacts.

Reviewer #3 (Remarks to the Author):

In this revised manuscript, the authors carefully addressed all of the reviewers suggestions. They identified and corrected several shortcomings in their original work, making this revision more accurate and technically sound.

I foresee no further comments, and I am satisfied with this version.